# Biodistribution of nanoplastics in mice: advancing analytical techniques using metal-doped plastics

Theresa Staufer [1,12] ✉, Verena Kopatz [2,3,4,5,12] ✉, Alice Pradel [6], Tess Brodie[7], Robert Kuhrwahl[1], Deborah Stroka [7], Julia Wallner[2], Lukas Kenner [2,4,5,8,9,10] ✉, Verena Pichler [4,11], Florian Grüner[1] & Denise M. Mitrano [6]

Contamination of plastic particles in environmental and biological systems raises concerns regarding their potential negative impacts. Human exposure to microplastics (MPs) and nanoplastics (NPs) is increasing; however, some potential adverse health effects might remain unclear, due to analytical challenges in detecting trace concentrations. To address this, we propose a workflow for NPs assessment in biological samples combining three complimentary methods, namely inductively coupled plasma mass spectrometry (ICP-MS), X-ray fluorescence imaging (XFI), and imaging mass cytometry (IMC) to detect palladium-doped NPs (Pd-NPs). This approach was used to quantify the temporal distribution and accumulation of Pd-NPs in mouse models under different experimental conditions, dosages, and time frames. Acute exposure showed a clear particle excretion from the gastrointestinal tract into feces, while subchronic exposure led to tissue accumulation. This workflow enhances our ability to analyze and study NP uptake and biodistribution mechanisms down to the nanoscale in complex biological samples.

The widespread distribution of plastics and their expected increasing production and use[1] have made humans vulnerable to exposure through various routes[2]. Plastic particles that persist in the environment can fragment into microplastics (MPs, <5 mm) and further into smaller nanoplastics (NPs, <1 μm)[1,3]. These contaminants have been found in seafood, milk, table salt, tap water, and bottled water[4,5], as well as in human stool[6], placental tissue[7], blood[8], and breast milk[9]. Even though this exposure has been observed for some time, the potential adverse health effects are not yet fully understood[2]. The most common micro- and nanoplastics (MNPs) uptake mechanisms for humans are ingestion and inhalation[9,10]. Particle size has a substantial influence on biological interactions and physicochemical behavior[11]. While it has been reported that MPs can translocate across biological membranes[12], there is limited evidence that micrometer-sized particles can penetrate in large quantities[13]. However, the biological fate for NPs is different, since they can more readily cross biological barriers[14,15] and

be taken up into cells[16]. They may also interfere with certain cellular functions such as metabolism, lysosomal and mitochondrial function[17], induce immune responses[18], or cause stress reactions (e.g., through induction of reactive oxygen species) and inflammation in tissues[2,19]. Adverse impacts to NP exposure in animal models and human cell lines, including CaCo-2, HCT116, A549, and THP-1, have been shown[10,20–22]. Consequently, the risk of internalization, accumulation, and adverse outcomes due to NPs exposure is a concern, but accurately quantifying NPs remains challenging in linking cause and effect[23,24].

There are dual analytical challenges with assessing NP burdens in any sample, including tissue extraction and quantification[24]. After sample collection, most analytical workflows extract and concentrate NPs, which rules out the possibility of spatial resolution. For quantification, the most commonly used tools to measure MPs in terms of particle number, including micro-Fourier transform infrared spectroscopy (μ-FTR), are generally not

[1]University of Hamburg, Fachbereich Physik, and Center for Free-Electron Laser Science (CFEL), Hamburg, Germany. [2]Department of Pathology, Medical University of Vienna, Vienna, Austria. [3]Department for Radiation Oncology, Medical University of Vienna, Vienna, Austria. [4]CBmed GmbH–Center for Biomarker Research in Medicine, Graz, Austria. [5]CCC–Comprehensive Cancer Center, Vienna, Austria. [6]Environmental Systems Science Department, ETH Zurich, Zürich, Switzerland. [7]Department of Visceral Surgery and Medicine, Inselspital, Bern University Hospital, University of Bern, Bern, Switzerland. [8]Unit of Laboratory Animal Pathology, University of Veterinary Medicine, Vienna, Austria. [9]Department of Molecular Biology, Umeå University, Umeå, Sweden. [10]Christian Doppler Laboratory for Applied Metabolomics, Medical University Vienna, Vienna, Austria. [11]Department of Pharmaceutical Sciences, Division of Pharmaceutical Chemistry, University of Vienna, Vienna, Austria. [12]These authors contributed equally: Theresa Staufer, Verena Kopatz. ✉e-mail: theresa.staufer@uni-hamburg.de; verena.kopatz@meduniwien.ac.at; lukas.kenner@meduniwien.ac.at

suitable for NP analysis as the size resolution is inadequate. Mass-based approaches, such as pyrolysis GC-MS, may therefore be more suitable for smaller-sized particles. However, the lengthy and involved workflow still needs development to accurately and precisely trace levels of NPs. While this calls for more sensitive and harmonized methods for measuring NPs, laboratory-based systems can take advantage of alternative approaches when using model particles. Labeling or doping NPs with rare elements[25], isotopes[26], or molecules[27] and using them as a tracer can circumvent many of the current analytical challenges associated with the NP analytical chain. Consequently, new avenues for (polymer-based) particle analysis become available.

When choosing model MNPs to study, one must keep in mind that MNPs are a large suite of contaminants with different properties such as polymer type, surface chemistry, extent of weathering, and others. These different physiochemical properties may also lead to variations in uptake and toxicological impacts, but the extent of this variation is not yet known and is currently under investigation by many scientists in the field. Therefore, we have chosen to use model nanoplastics with defined properties administered as a homogeneous suspension. The model metal-doped materials are a good proxy for environmental nanoplastics due to their surface composition and irregular surface. A number of different doping techniques have been put forth in recent years, including the addition of fluorescent dyes[28], inorganic metal-doping[11,29], radio-labeling or stable-isotope labeling[26] of the carbon polymer backbone. The commercial availability and ease of use of fluorescently labeled model MNPs have made then a popular choice amongst researchers, but challenges associated with leaching of dyes, as well as quenching of the fluorescence with organic matter, have caused concern in the past. Radio or stable isotope labeling offers excellent detection limits, but the analytical equipment needed in order to measure the materials is often not commonplace. Consequently, metal-doping has several key advantages, since existing standard methods for trace metal analysis exist and can be exploited for measuring metal-doped plastic materials[23,30].

Here, we show a comparison of three different analytical methods to quantify metal-doped model NPs in biological samples, namely inductively coupled plasma mass spectrometry (ICP-MS), X-ray fluorescence imaging (XFI), and imaging mass cytometry (IMC). ICP-MS has proven to be a useful approach either for the direct measurement of metal-doped NPs (e.g., through single particle ICP-MS) or microwave acid digestion followed by bulk metals analysis[31–33]. In contrast, XFI and IMC have not been utilized in this context before. ICP-MS can quantify metals and metalloids in trace concentrations in liquid, gaseous, and solid matrices[34]. In this study, samples were analyzed in liquid form, which required solubilizing samples by digestion. Therefore, this analytical technique is destructive and the spatial distribution of metals within a sample is lost. XFI is a novel imaging method based on the detection of characteristic photons in the hard X-ray range after excitation with a scanning pencil X-ray beam[35–40]. Medium- to high-Z elements are required as markers for XFI, for example, palladium or gold nanoparticles[41] or molecular markers such as iodine atoms[42]. The spatial resolution achievable in XFI is only determined by the applied X-ray beam diameter[35]. A clear advantage of XFI over the commonly used optical fluorescence imaging methods for MNPs is that the emitted characteristic X-ray photons are energetic enough to penetrate large distances through tissue, even up to the human scale[36,43]. Thus, XFI allows for the detection of model metal-doped NPs in complex biological matrices and in vivo, and provides quantitative information in combination with spatial resolution. Due to the non-invasive nature of the method, samples can subsequently be analyzed with other techniques, as presented in our study. IMC is an innovative technology[44] that combines the sensitivity of mass spectrometry with spatial histology[45]. Tissue is typically stained with antibodies coupled to heavy metals to identify cellular proteins or, as in this study, when the animal is gavaged with metal-doped NPs. Sliced sections are subsequently UV laser ablated, and the released material is analyzed in a Helios mass cytometer. Inside the Helios, the material is atomized and ionized, filtered, and pushed into the time of flight (TOF) chamber for separation of masses and their

subsequent detection. The $x$ and $y$ positions of the ions are recorded, so the data includes both spatial and intensity information from each metal detected.

To quantify the temporal distribution and accumulation of polystyrene (PS) NPs in model mouse systems, metal (palladium)-doped PS NPs (Pd-NPs) were applied in three different exposure regimes (acute, subacute, and subchronic) covering different biological conditions and different time frames. This included healthy mouse models of Pd-NPs exposure as well as disease models of colitis for subacute exposure and APCmin+ (intestinal polyps) at subchronic exposure to also cover the influence of certain intestinal diseases on NP uptake and tissue accumulation (see schematic in Fig. 1). Colitis is an inflammatory process, which mainly affects the colon and causes a leaky intestinal barrier which is supposed to lead to higher MNP uptake rates into the body[46]. The APCmin+ model, in contrast, is a genetic model of early-stage intestinal neoplasia, with benign adenomatous polyps mainly forming in the small intestine and to a lesser extent in the colon[47,48]. The model is considered similar to human familial adenomatous polyposis and also shows to be associated with a defective intestinal barrier and increased permeability[49]. As the gastrointestinal (GI) tract is chronically exposed to orally ingested MNPs and—aside from inhalation—is probably one of the main entrance routes for MNPs into the body, we also wanted to investigate Pd-NP uptake under these pathological conditions. As we set out for this multifactorial approach to be able to go for both optimal quantification and spatial resolution, we first used non-destructive XFI scans to get an initial impression of the spatial Pd-NPs distribution in the sample, with a typical resolution in the mm-range. Such scans do not require any special sample preparation and provide a first quantitative impression of the dynamics of Pd-NPs, however, the achievable sensitivity is lower compared to the other methods. Based on these results, ICP-MS and IMC measurements were conducted, providing a higher detection sensitivity and a resolution down to the cellular level. While the analytical workflow of ICP-MS could measure a larger number of samples, IMC required a good knowledge of regions of interest a-priori, as the laser ablation process takes time (~ 2 h per 1 mm$^2$). Hence, a combination of all three methods was advantageous to determine the translocation and potential accumulation of Pd-NPs in terms of minimizing time investment while optimizing for spatial distribution and high sensitivity. Most samples were measured with ICP-MS and XFI and selected ones were also measured with IMC to confirm the detection of Pd signals with much higher spatial resolution down to the cellular level. Collectively, this work presents a direct comparison of promising analytical methods to assess the biodistribution of model metal-doped PS NPs in tissues while at the same time providing insights into the kinetics of NP uptake and distribution in an organism at different exposure scenarios and also disease models.

## Results
### Concentration of model metal-doped nanoplastics in the gavage solutions

Three different concentrations of Pd-NPs were used for the experiments. In the pilot acute exposure scenario, we used the undiluted stock concentration. Although this concentration is high, we needed this high dose of particles to test if we were able to detect the particles with the different analysis methods and to also track the NP uptake into the body, as we already expected uptake amounts to be very low after this short exposure period. For the subacute and subchronic exposure, we reduced the applied daily dose to more physiological concentrations of 1 mg and 0.5 mg/day.

The measured concentration of the gavage solution of the acute exposure experiment (i.e., particle stock) was 99.76 µg Pd/mL, as measured on average by all three analytical approaches (detailed values in SI). For XFI, the suspension was measured directly, and for ICP-MS and IMC, the solution was measured after serial dilutions in deionized water. Mice were given a single dose of 100 µL in the acute exposure experiment, which equated to a total dose of 9.98 µg Pd (3.38 mg NPs). For the subacute and subchronic exposure experiments, the daily doses were 2.91 µg Pd (0.99 mg NPs) measured by XFI and 1.41 µg Pd (0.48 mg NPs) measured by ICP-MS,

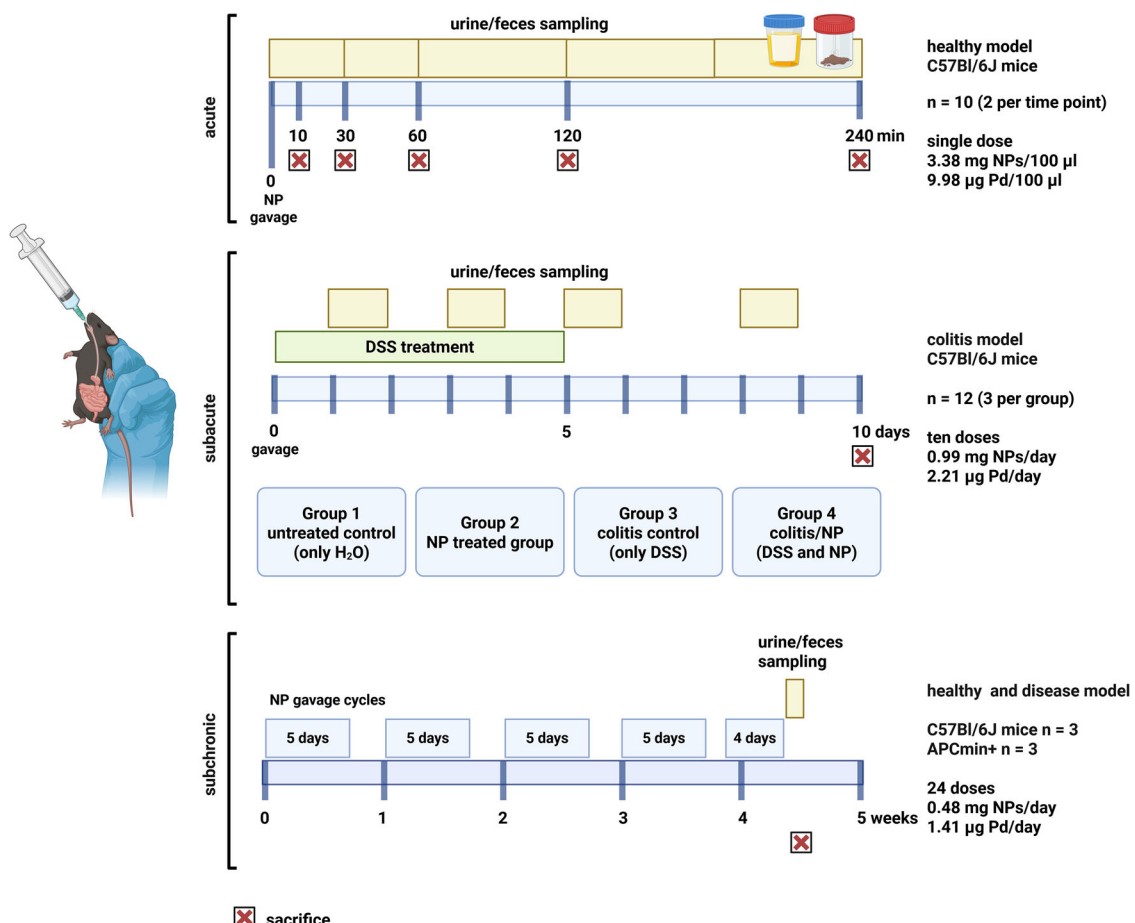

**Fig. 1 | Mice treatment scheme for acute, subacute and subchronic Pd-NP exposure.** Timeline for gavage and sampling (Created in BioRender. Pichler, V. (2025)). In the acute exposure experiment, 10 mice received a single gavage of Pd-NPs (3.3817 mg NPs; 9.976 μg Pd) and were processed at 5 time points post-gavage: 10, 30, 60, 120, and 240 min while urine and stool were also sampled at distinct intervals up to 240 min. The subacute exposure experiment was conducted in a healthy as well as a colitis-induced model (4 treatment groups, $n = 3$) with gavage for 10 days (0.99 mg NPs; 2.91 μg Pd daily dose). Stool was collected over 24 h at 4 time points and all mice were sacrificed 24 h after the last gavage. For the subchronic exposure experiment, three healthy and three APCmin+ mice were gavaged for 5 weeks (0.48 mg NPs; 1.41 μg Pd daily dose) and feces and urine were collected for 24 h after the last gavage dose.

respectively. These values were used for further data analysis (i.e., Pd content as a function of total or daily dose).

## Acute exposure experiment — Pd-NPs are mainly found in the lumen of the digestive tract with minimal translocation

The first experimental approach was designed to assess how quickly Pd-NPs move through the GI tract of a healthy mouse, whether the particles could cross the intestinal barrier, and if they were taken up into the organs. Healthy wildtype mice were treated with a single high dose of Pd-NPs (3.38 mg NPs) and examined after short exposure durations (10 min—4 h). XFI measurements revealed Pd-NPs in the GI tract and the feces, while ICP-MS also detected signals in other organs. Additional IMC measurements confirmed Pd-NPs signals in selected organs. An overview of all analyzed samples for all 3 experimental settings and across all 3 analysis methods is presented in Fig. 2.

The trajectory of Pd-NPs through the GI tract was elucidated through a series of XFI-maps, illustrating the spatial and temporal dynamics of Pd-NPs transport through the small intestine from the outer (proximal) to the inner (distal) end (Fig. 3a, see Fig. S1 for a combination of XFI mass maps and optical images of the tissue blocks). Quantification of Pd distributions from the XFI-maps of GI organs (stomach, small intestine, cecum, colon; measured with content) showed a clear excretion pathway from the stomach into the subsequent organs of the GI tract in a timely manner, as also measured by ICP-MS (Fig. 3b). The total mass of Pd-NPs measured in the GI tract decreased over time as more Pd-NPs were either excreted *via* feces or potentially taken up from the GI tract and translocated into other organs. Of note, one XFI measured datapoint for the total recalculated Pd content in the GI tract was measured to be >100% of the applied dose at the 30 min time point (125.62%), yet the majority of datapoints lie within the expected range. The excretion of particles from the body was also measured through the fecal Pd content (time course, pooled from two individuals, Fig. 3b and Table S1). Pd in the feces was detected with XFI as early as 2 h post-gavage (0.03% of the applied dose) and increased until 4 h up to 30.23%, assuming that both individuals contributed equally to the mass of Pd excreted *via* the feces. Further information and a summary of the scanned samples, reconstructed Pd masses, and detection limits in the current XFI-setup for a representative mouse for the acute exposure experiment can be found in Table S1. Aside from the GI organs and feces, no Pd-NP signals were detected with XFI in other organs or blood.

With ICP-MS, the highest concentrations of Pd were detected in the different GI tract organs (stomach, small intestine, cecum, colon; measured with content) and in the feces, thus again highlighting the movement of the Pd-NPs over time through the GI tract (Fig. 3b and Table S2). While low concentrations of Pd-NPs were present 2 h after administration (0.09%) in the stool samples, most Pd-NPs were excreted 3 to 4 h post-gavage (30.69%). Pd was also detected in five of the ten pancreas and one blood sample (sampled 30 min after gavage). No Pd was detected above the detection limit (< 0.42 ng) in the other measured organs (brain, liver, mesothelium, heart, spleen, kidney, and testes). The complete data set on the total Pd concentrations in all organ samples, unflushed GI tract and feces, as well as the

**Fig. 2 | Overview of the organs investigated for the acute, subacute, and subchronic exposure experiments.** (Created in BioRender. Pichler, V. (2025) https://BioRender.com/v18f373). Shown are all organs which have been analyzed in the three experimental setups. The different colors indicate the analytical method and the symbols show whether Pd-NPs were (partly) detected.

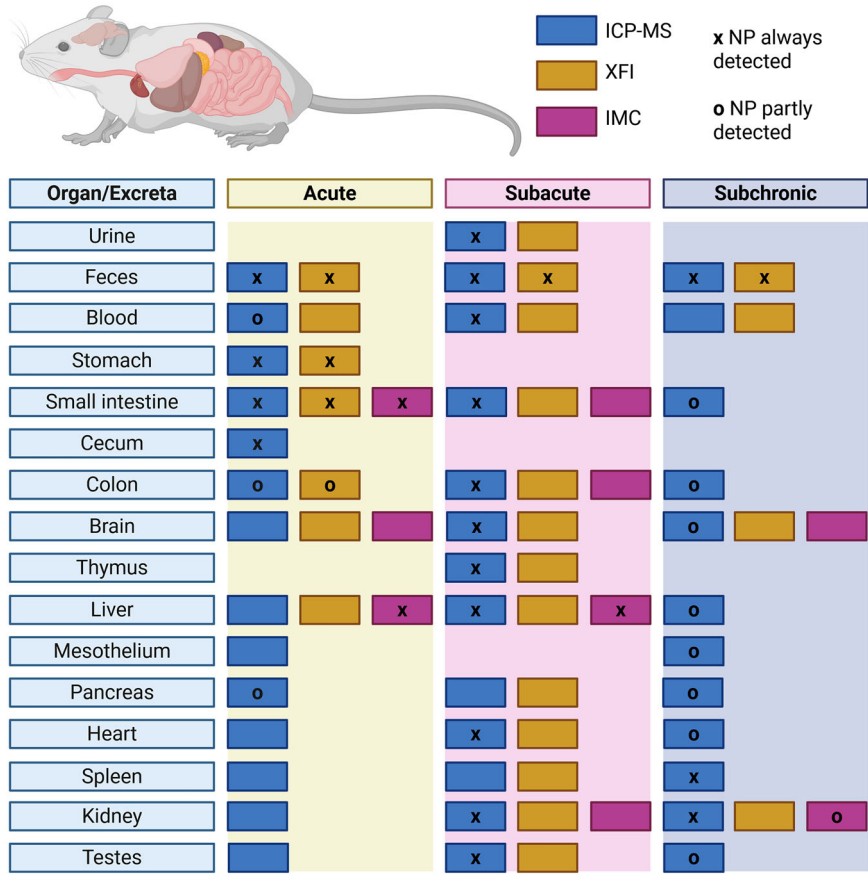

respective recalculated percent of daily dose ( = applied dose), can be found in Tables S2–4.

To test the feasibility of detecting Pd-NPs by IMC, samples from the brain, liver, and small intestine (without stool removal) were chosen. Samples were selected based on previous XFI and ICP-MS measurements and whether Pd particles could be detected in this sample (small intestine as a positive sample, brain and liver as potentially negative samples, see Fig. S2A). Precise signals for Pd-NPs were detected in the small intestine (yellow spots in Figs. 3c and S2B, dual counts above 1.5 in Fig. 3d). In the liver, one pixel was detected at 2.4 dual counts, while no signals were found in the brain (Figs. 3d and S2C, D). The Pd signals were mainly seen in the feces of the small intestine, more specifically in contact with and in between the intestinal villi (Figs. 3c and S2B). To quantify the Pd signal from the tissue images, pixel counts graphs were created to report the number of detected $^{108}$Pd ion counts per pixel in each ROI (Fig. 3d). $^{108}$Pd was selected because out of all the naturally occurring palladium isotopes present, $^{106}$Pd and $^{108}$Pd had the best detection sensitivities as shown in Fig. 3e. An overall background signal of 0 to 1 dual count per ROI was detected (Fig. 3d). Above this, in the small intestine there was a gradation of Pd counts up to a maximum of 6 dual counts. Since 1 pixel is 1 µm, and the NPs were approximately 200 nm in diameter, it is anticipated that 1–5 Pd-NPs can fit into the space of one pixel, explaining the variability in the signal strength between 1.5 and 6 dual counts per pixel. Five out of eleven ROIs in the small intestine had Pd signals above the noise, and they were all in regions containing feces. In the liver, one pixel in one ROI out of 4 had detectable levels of Pd.

In the acute exposure experiment, a precise tracking of the Pd-NPs along the GI tract and quantification of excretion in the feces was possible. Overall, total Pd signals in the GI tract were around—or close to −100% of the applied dose within the first 60 min after application and decreased to 80% at 120 min and 40–50% after 4 h. Pd signals in the stomach and small intestine were strongest between the first 60 min and declined afterwards.

Signals in the colon and cecum began to increase after 120 min, however, increases for the colon were not as considerable and reached approximately 20% of the applied dose after 4 h. Traces of Pd were detectable as early as 2 h post-gavage in the feces (XFI: 0.03% or ICP-MS: 0.09% of the applied daily dose), suggesting a fast transition of a very small portion of Pd-NPs. Most of the particles, however, passed the GI tract at a much slower rate. Approximately 30% of the applied dose was excreted between 3 and 4 h post-gavage, thus being in line with the decreased total Pd content in the GI tract (~ 40–50%). In summary, during the first 4 h post-gavage, ~80% (with XFI for the earlier time points also up to 100%) of the total Pd-NPs applied could be tracked and located along the GI tract (majority in the content of the GI tract lumen) or in the feces.

## Subacute exposure experiment—Pd-NPs are taken up into distal organs

While in the acute exposure experiment, rather high particle concentrations and short time points were studied, the Pd-NPs dosage was reduced to more physiologically relevant doses (0.99 mg NPs) in the subacute exposure experiment and the time points extended up to ten days. In addition, a healthy mouse model was compared to a dextran sulfate sodium (DSS) induced colitis model to examine whether a harmed intestinal barrier influenced the uptake and translocation of Pd-NPs. Assuming that inflammation triggers a higher intestinal barrier permeability, we tested if NPs could "leak out" of the gut in this model to a greater extent.

While several organs were measured with XFI, notably, Pd was only detected in the stool samples with this method (Table S5). In the groups undergoing DSS treatment, feces sampling was not possible due to the individuals having diarrhea. Thus, only feces from the control and Pd-NP-treated groups were sampled. Stool samples were collected over 24 h post-gavage at various time points over the course of the experiment. Figure 4a displays an exemplary Pd XFI-map of feces collected on day 9. The Pd

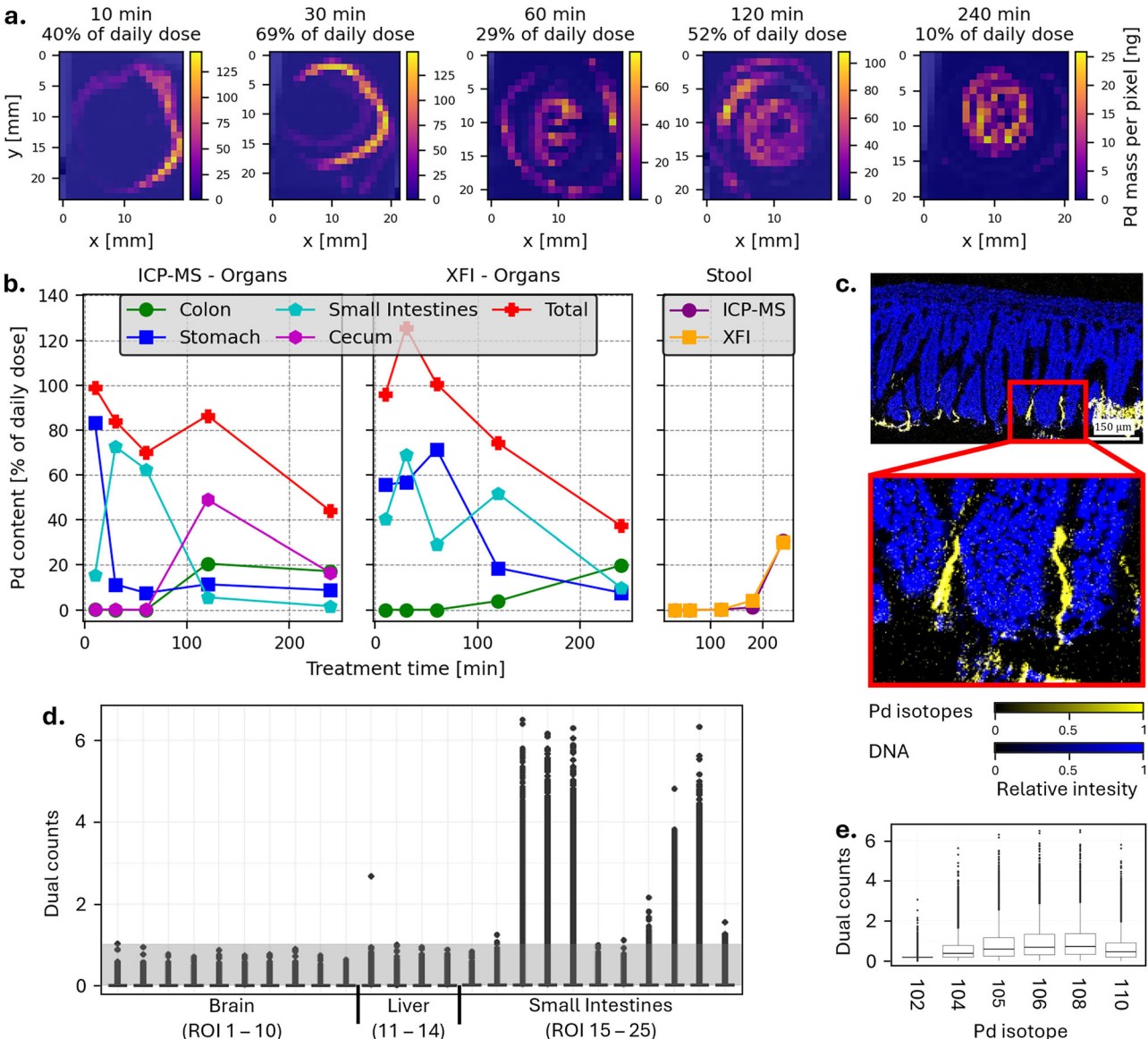

**Fig. 3 | Acute distribution and excretion of Pd-NPs in healthy mice. a** XFI-maps depicting the translocation of Pd-NPs in the small intestine over time. **b** Absolute Pd masses in the GI tract of one mouse measured with ICP-MS (left panel) and XFI (middle panel) as a function of the daily dose ( = applied dose), as well as averaged Pd stool mass from two mice (right panel). ICP-MS and XFI measured organs from different mice. **c** IMC tissue image of a small intestine showing $^{104}$Pd, $^{105}$Pd, $^{106}$Pd, $^{108}$Pd, $^{110}$Pd in yellow and DNA in blue for region of interest (ROI) si13 (small intestine from mouse #28671, 240 min). **d** IMC dual counts per pixel for $^{108}$Pd for 10 ROI in the brain (1–10), 4 ROI in the liver (11–14), and 11 ROI in the small intestine (15–25) of mouse #28671 (240 min). The grey box between 0 and 1 dual counts indicates the noise range of the detector and only pixels above that area reliably contain Pd. Respective ROI images are shown in Fig. S2. **e** IMC of all Pd isotopes expressed per pixel in small intestine ROI 3 (mouse #28671). The box plots display the first (upper) and third (lower) quartiles around the median, with the black line in the middle of the box indicating the actual median value. All points that fall outside this range are indicated as dots above and below the boxes.

signals can be clearly correlated with the feces in the sample tube, which were collected from 3 individuals. The total reconstructed mass of roughly 9.0 µg Pd shows a high clearance of the administered Pd-NPs of about 3 daily doses (104% per mouse).

Through ICP-MS analysis, Pd-NPs were mainly detected in feces samples from both treatment groups and, to a lesser extent, in a few organs of one mouse from the Pd-NPs+DSS treatment group. Trace concentrations of Pd were also measured in untreated control feces for days 1 and 3 (2.31% and 0.57% of the daily dose, respectively) but declined in subsequent time points. However, the Pd content in the Pd-NPs treated group was higher at all time points, starting from 3.89% of the daily dose on day 1 and constantly increasing to 82.72% of the daily dose on day 9. A detailed timeline of Pd-NPs content in the feces is shown in Fig. 4b and Table S6. For the analysis of mouse organs, due to the very low detectable Pd measured in the Pd-NPs

treatment group by XFI, we only measured organs from one Pd-NPs+DSS treated mouse with ICP-MS (Table S7). Detectable concentrations of Pd were measured in the organs of the GI tract (small intestine 32.80% and colon 2.83% of the daily dose, flushed organs). However, the relatively high concentrations measured in the small intestine of this one individual should be considered with caution, as remnants of stool from the GI tract could also have been present even after flushing. Furthermore, Pd was also detected in other organs, including the thymus (0.82%), liver (1.24%), and testes (0.20%). No measurable Pd concentrations were detected in the pancreas or spleen. We anticipated that Pd concentrations in the Pd-NPs+DSS treated mice would be higher than the Pd-NPs treatment alone because more particles might be able to cross a non-intact intestinal barrier.

Based on the quantification of Pd from the ICP-MS analysis, colon, small intestine (washed out), liver, and kidney were selected for IMC

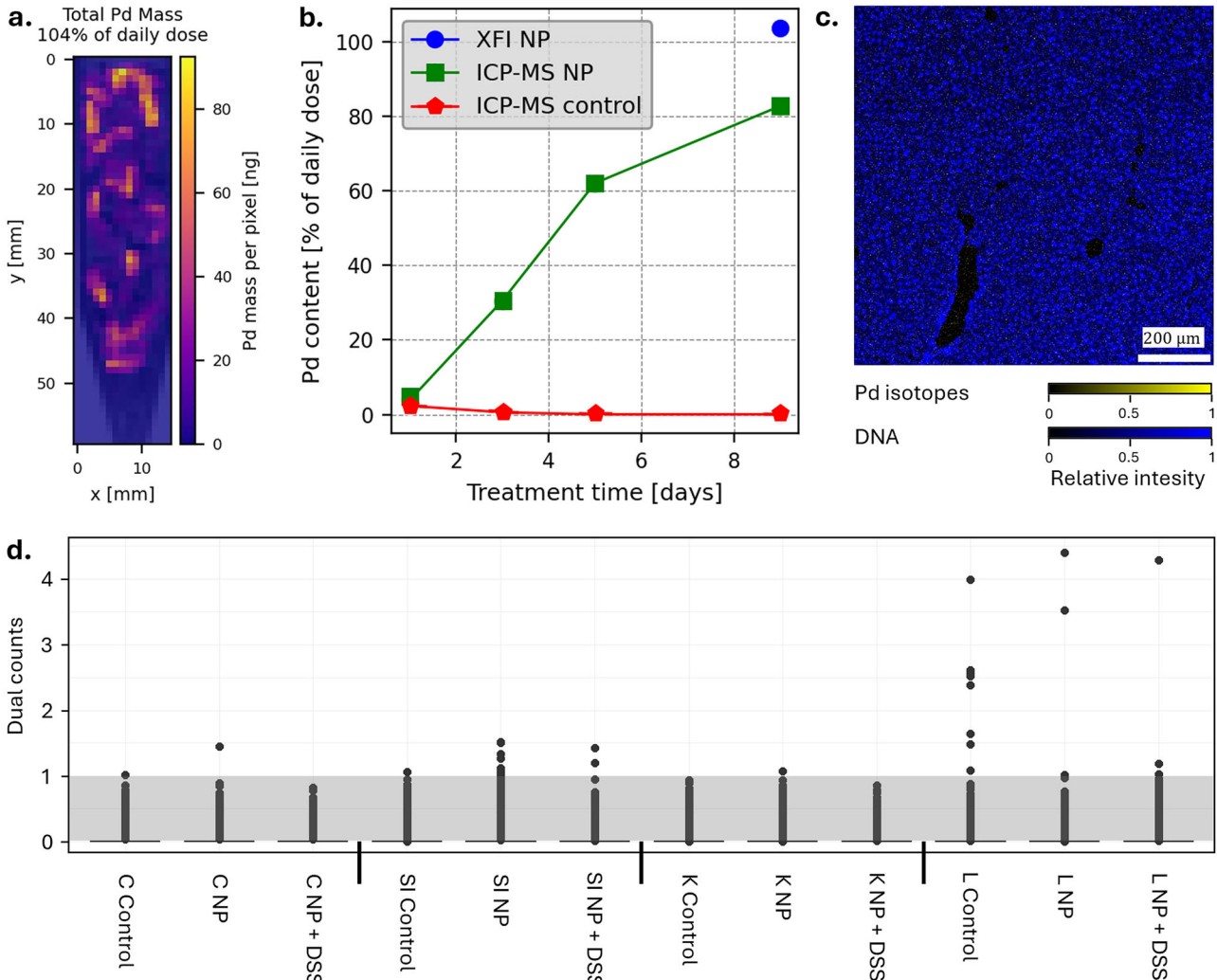

**Fig. 4 | Subacute distribution and excretion of Pd-NPs in healthy and DSS mice.**
**a** XFI-maps showing the Pd content in the mouse droppings collected over 24 h after Pd-NPs gavage on day 9 (content from 3 mice, recovery of Pd correlates to 104% of the daily dose). **b** Absolute Pd masses in excreted feces (recalculated as % daily dose/ mouse) measured with ICP-MS (green) and XFI (blue). Mice that did not receive Pd-NPs gavage (untreated control) are displayed in red. **c** IMC tissue image of liver showing $^{104}$Pd, $^{105}$Pd, $^{106}$Pd, $^{108}$Pd, $^{110}$Pd in yellow and DNA in blue for the liver sample from Pd-NPs+DSS (mouse #28334). **d** IMC dual counts of $^{108}$Pd in the colon, small intestine, liver, and kidney for the 3 treatments (untreated control, NP, and Pd-NPs+DSS) (L Liver, K Kidney, C Colon, SI Small Intestine) in 3 mice (untreated control = mouse #28328, Pd-NPs = mouse #28337, Pd-NPs+DSS = mouse #28334). The grey box between 0 and 1 dual counts indicates the noise range of the detector and only pixels above that area contain Pd. Respective ROI images are shown in Fig. S3.

analysis. We analyzed untreated control, Pd-NPs, and Pd-NPs+DSS samples for comparison. Detectable but low signals were measured in liver samples from the untreated control, Pd-NPs, and Pd-NPs+DSS treatments, but hardly any in the colon, small intestine, or kidneys (Figs. 4d and S3). The liver of the untreated control mouse had 4 pixels with Pd signals above background (> 2 dual counts), the Pd-NPs mouse had 2, and the Pd-NPs+DSS mouse had 1 pixel. As in Fig. 4c, the reader will notice that the yellow pixels containing Pd are impossible to see by eye because they are mixed with the background noise of the detector. They are shown to provide an example of what it looks like when we have extremely few positive pixels for Pd. For this reason, we use the pixel counts dot plots with dual counts to show that these images do in fact, contain a couple of pixels with Pd in them (signals above the background of 1 dual count). Each spot in the pixel counts dot plots represents one pixel and the $y$-axis provides the intensity of the Pd signal for that specific pixel in dual counts. As most pixels had no Pd detected, they have a signal intensity between 0 and 1 (in the grey range), which is the noise range of the detector. The maximum intensity of the pixels was between 4 and 4.5 dual counts.

The subacute exposure experiment gave hints of potential Pd-NPs accumulation already in various organs of Pd-NPs+DSS treated mice (liver, thymus, and testes; > 0.5% of the daily dose), as well as from the GI tract, after 10 days of particle application. Although Pd levels were relatively low (< 1% of the daily dose), these findings show that Pd-NPs are taken up from the GI tract into the organism, penetrate distal organs, and might accumulate over longer timescales. Overall, the Pd content of feces collected for 24 h at various time points after gavage also showed an increasing trend over time from ICP-MS analysis, starting from rather low concentrations of less than 4% of the daily dose on day 1. However, we suggest that these low values in the collected feces might instead be due to an inhomogeneous Pd distribution within the sample (as seen with XFI-analysis of the whole sample) and a problem of subsampling for the 24 h stool collections, as ICP-MS could not analyze the entire sample. As observed during the acute exposure experiment, in which only a single or very few stool droppings had to be homogenized, the excretion values were already rising to ~30% of the daily dose after 4 h. Also, Pd values for samples collected later during the experiment on day 9 were concordant between the two analytical methods. Both ICP-MS and XFI measurements showed that Pd-NPs of about one

daily dose (82.7% or 104% of the daily dose) were excreted over a time course of 24 h.

## Subchronic exposure experiment—Pd-NPs accumulate in distal organs over time

To study the potential subchronic accumulation of Pd-NPs in healthy as well as diseased mice (APCmin+, genetic model of GI polyps), lower concentrations of Pd-NPs (0.48 mg NPs) were used, and mice were gavaged over a period of ~5 weeks. Additionally, feces were taken at hourly intervals after the last gavage, thus complementing the data from the acute exposure experiment with a longer observation timeframe up to 10 h post-gavage. This experimental design could verify the results from the subacute exposure experiment, showing that Pd-NPs could be taken up from the GI tract into the organism and reach distal organs.

XFI measured considerable Pd signals in a time series of stool samples, following Pd-NPs excretion for 10 h post-gavage (Fig. 5a, b). XFI-maps of the individual droppings from wildtype mice sampled per time point (Fig. 5a) showed a similar excretion pattern as in the acute exposure experiment. First signals of Pd-NPs in the feces were visible as early as 2 h post-gavage. They increased with time, exhibiting a maximum excretion of 12–13% of the daily dose being excreted at the 4 and 5 h time points after gavage, before levels began to decrease (Fig. 5b). Summing all Pd values in the feces collected during the time course, wildtype mice were shown to excrete up to 46% of the daily dose during the first 10 h after Pd-NPs administration. Concentrations of Pd in other organs or blood could not be detected with XFI (see summary in Table S8).

Stool samples from both wildtype and APCmin+ mice were measured with ICP-MS (Fig. 5b). While wildtype mice showed a more rapid transition of Pd-NPs with the maximum particle excretion between 3 (9%) and 4 (6.09%) with decreasing Pd-NP concentration after ~5 h, APCmin+ mice had a much slower increase in Pd concentrations in the feces. Here, the highest Pd values in the feces occurred at 6 (5.87%) and 9 h (9.42%), with a break between 7 and 8 h. Notably, feces droppings from mice during the 6–8 h post-gavage time frame were very limited in both treatment groups, with sometimes no or only one dropping at a time point for the entire group. After 10 h, both treatment groups had excreted ~30% of the daily dose administered. The Pd content in terms of % daily dose recovery for each time point and a graphic representation of the cumulative Pd dose are shown in Fig. 5b and detailed data is depicted in Table S9.

Overall good agreement in terms of Pd content in % of daily dose between the methods of XFI and ICP-MS can be found for the majority of data points, while lower values were observed for the 4 and 5 h time point using ICP-MS. Such lower values can be understood by assessing the sample preparation process for ICP-MS and the sample material. While in XFI, the whole samples were scanned without modification, ICP-MS required transfer of the feces into digestion tubes. Thus, especially for soft feces samples, it is possible that parts of the material were stuck in the sampling tube and not quantitatively transferred. Therefore, lower Pd masses might be found for feces samples with ICP-MS compared to XFI. However, regular organ measurements with ICP-MS should not be affected by this issue.

Pd was detected with ICP-MS in the brain, liver, heart, and kidney of the subchronically exposed mice (average organ Pd content from all mice > 1% of the daily dose, see Fig. 5c). However, some concentrations measured were below the limit of detection (LOD) or limit of quantification (LOQ). Almost all liver and kidney samples (4 out of 5 (4/5)) had Pd levels above > 1% of the daily dose. In some mice, there were selected organs with significantly increased Pd concentrations. For example, Pd levels in one liver sample from an APCmin+ mouse (#29200) were significantly elevated, with 44.65% of the daily dose. Additionally, one sample from the brain (in an APCmin+ mouse) and one from the kidney (in a wildtype mouse) reached levels of > 10% of the daily dose. Lower Pd signals were detected only in some mice in the brain (2/5 > 1% of the daily dose), mesothelium (both measured samples > 0.5%), pancreas (2/5 > 1%), heart (3/5 > 0.5%), spleen (2/5 > 0.5%) and testes (3/5 > 1%). No Pd was measured in the blood of both treatment groups. Total Pd recovery rates for all organs per mouse were

24.68% and 10.61% of the daily dose for the two wildtype mice and 48.30, 16.93, and 5.57% of the daily dose for the three APCmin+ mice (see Tables S10 and S11). Although single APCmin+ mice exhibited quite high Pd signals in distinct organs (e.g., brain, liver), the overall Pd levels per organ were comparable to those of wildtype mice, with the exceptions of heart and spleen. In heart samples, surprisingly higher Pd concentrations were detected in wildtype mice (wildtype 2.81% vs APCmin+ 0.34% of daily dose, $p = 0.073$), whereas for spleen samples, all three APCmin+ mice had higher Pd concentrations (wildtype 0.19% vs APCmin+ 0.87, $p = 0.122$). However, both findings could not reach statistical significance with the low number of animals used in this study. Also, total Pd recovery rates per mouse were observed to be in a comparable range between wildtype and APCmin+ mice. Lastly, we assessed whether a subchronic exposure to Pd-NPs was detectable in tissues using IMC. Positive signals were found in some brain and kidney samples with ICP-MS, so these tissues were selected for more detailed investigations. The kidney from wildtype mouse #29194 had 2 of 3 ROIs with signals above background (Figs. 5c, d and S4) and no signals were measured in the kidney from mouse #29207 (APCmin + ) or in the brain from mouse #29194 (wildtype).

Subchronic exposure experiments in healthy and APCmin+ mice verified the results from the subacute exposure experiments, that orally applied Pd-NPs could be taken up from the GI tract and reach distal organs. Increased Pd concentrations were similarly found in liver and kidney samples (4 out of 5 samples had values > 1% of the daily dose), but also in some brain, heart, and testes samples and to a lower extent mesothelium, pancreas, and spleen samples. Prolonged feces sampling after the final gavage showed that healthy wildtype mice excrete Pd-NPs rather quickly, with most particles being excreted between 3 and 5 h after application. In contrast, diseased APCmin+ mice, having various intestinal polyps/lesions in the small intestine, showed a slowed/delayed transition time of the particles. Although cumulative Pd levels after 10 h showed similar concentrations of Pd (~ 30% of the daily dose for ICP-MS or 46% of the daily dose for XFI measurements) to be excreted until that point, a slower transition of Pd-NPs through the GI tract might bear a greater potential for interaction, uptake, and adverse effects.

## Correlation and opportunities for advanced analytics in NPs biological distribution studies

Each analytical tool presented in this study could accurately quantify model metal-doped Pd-NPs within biological tissues, allowing us to assess the biokinetics and distribution of Pd-NPs in mice. In principle, the results correlated well and together provided complementary data with different sensitivity levels and possibility for spatial resolution. While ICP-MS had the lowest absolute detection limit and was the technique that was capable of the highest throughput, spatial resolution was lost, and Pd-NP burdens could only be measured on an organ level. In contrast, both XFI and IMC had superior spatial resolution, but the time investments to perform the analysis and access to instrumentation (especially in the case of XFI measurements at a synchrotron) may make the application of the technique(s) limiting to some researchers. To overcome the latter issue, compact, that is, laboratory-scaled X-ray systems capable of performing high-sensitivity XFI measurements are currently under development at the University of Hamburg with a prototype already in use for first test measurements. While IMC had high sensitivities and excellent possibilities to assess precise localization of Pd-NPs, without some a-priori information, e.g., from XFI, on which samples to test, scanning entire organs seems impractical due to long laser ablation times. Collectively, these initial results are encouraging, and it would be worthwhile to perform longer-duration experiments (e.g., 6 months) at varying Pd-NPs doses to assess how prolonged, subchronic exposures may allow for the accumulation of NPs in distal organs.

There were some discrepancies between XFI and ICP-MS measurements, even for the same sample, but these could be (at least partially) explained by means of different sample preparation needs. As could be seen in Fig. 4a, Pd signals were heterogeneously distributed in the stool droppings over the 24 h sampling period. As the ICP-MS workflow required

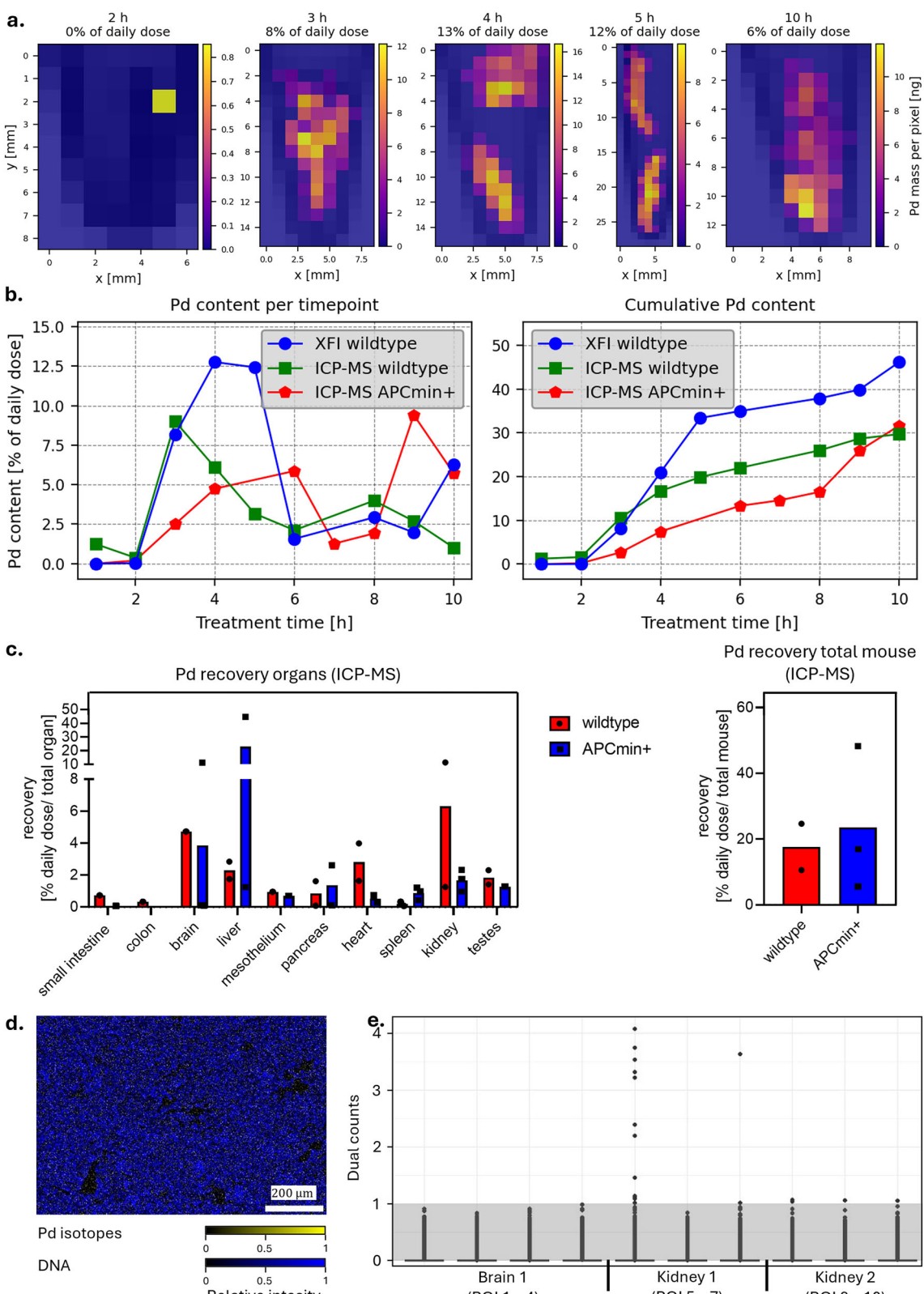

**Fig. 5 | Subchronic distribution and excretion of Pd-NPs in healthy wildtype and APCmin+ mice. a** XFI-maps showing the Pd-NPs in the feces droppings excreted over time in two wildtype mice. **b** Pd masses as measured by XFI and ICP-MS given as percentage of the daily dose per mouse as the excreted masses per time point (left) and the cumulative mass (right). ICP-MS and XFI measured the same samples. **c** Pd-NP recovery rates. ICP-MS measurements were recalculated for total organs (left) and summed up for total mouse (right) and given as % of daily dose. Data points represent individual measures, bars indicate mean values per group. **d** Tissue image of a kidney showing $^{104}$Pd, $^{105}$Pd, $^{106}$Pd, $^{108}$Pd, $^{110}$Pd in yellow and DNA in blue for ROI kidney194_1_1 (kidney from mouse #29194/wildtype). **e** IMC dual counts per pixel in brain and kidney showing $^{108}$Pd for 2 mice (1 = mouse #29194/wildtype, 2 = mouse #29207/APCmin+). The grey box between 0 and 1 dual counts indicates the noise range of the detector and only pixels above that area contain Pd. Respective ROI images are shown in Fig. S4.

homogenization and subsampling steps, this could be prone to errors and consequently discrepancies in the measurements could arise. In contrast, sample preparation of the organ halves for ICP-MS measurements only required drying of the native organs and transferring the samples from transport tubes to digestion tubes, to assure for a minimal loss of Pd-NPs from sampling to analysis. For XFI and IMC measurements, however, the organs were fixed and desiccated (formalin-fixed paraffin-embedded, FFPE), requiring several washes in different solutions, which may lead to washing out some of the particles. Although XFI would be also able to measure Pd-NP content in dried organs (similar to ICP-MS, without sample processing to avoid potential particle loss), we proposed the workflow with paraffin embedding for a better correlation with IMC analysis and to further enable subsequent immunohistochemical analysis if desired. Chemical compatibility of polystyrene particles with fixation and a desiccation protocol was already described[50] and independently verified a priori. Furthermore, good correlations between values obtained by ICP-MS, where samples were only dried, and by XFI, where samples were processed with FFPE, especially for acute exposure samples, highlight the negligible losses of Pd-NPs during tissue FFPE processing.

The IMC data from this study reports, for the first time, that NPs in the range of 200 nm can be detected with this technology at a sensitivity of one particle correlating to one dual count. For low-intensity signals (below 30 dual counts), the number of dual counts is the number of ions hitting the detector, so that we can correlate one NP to one dual count to one ion hitting the detector. In the case of these model Pd-NPs, there is some slight variability in the metal loading per particle[16] and thus, there is also some variability in the measured dual counts per NP. This work confirms that Pd-NPs can be used to study the cellular distribution of metal-doped NPs in animal models and that, since the signals are low alongside the time investment to measure samples, it is best to use this technology in parallel with technologies that can first detect which organ(s) to measure, such as ICP-MS or XFI. This work further supports the concept of performing future studies with IMC using antibodies that can identify the cell types present and reveal which cell subsets and neighborhoods harbor the NPs. To improve the sensitivity with IMC, other metals that are in the highest ion detection mass range for IMC (153 to 176) could be used or particles with a higher metal loading could be synthesized.

A detailed comparison of the analytical merits of the three methods employed in this study can be found in Supplementary Table S12.

## Discussion

We have systematically studied the uptake and translocation of model Pd-NPs in mice for different exposure conditions using three complimentary analysis methods. Acute exposure to high particle doses showed a clear excretion pathway along the GI tract lumen into feces, which all three methods could confirm. ICP-MS and IMC were sensitive enough to detect particles in organs other than the digestive system, such as pancreas, liver, kidney, and blood. While only healthy mice were studied for the acute exposure, a colitis model was used for the subacute exposure experiment with a lower dose of particles. XFI did not detect particles in samples other than feces. However, ICP-MS revealed particle concentrations in the flushed small intestine and colon as well as in the liver for DSS-treated mice. IMC also detected particles in the liver. In the subchronic exposure study, the particle exposure was again reduced along with the introduction of a genetic disease model for intestinal polyps (APCmin+). Similar to the previous results, most signals were detected in the feces, with a maximum at approximately 4 h post-gavage. No increased uptake by the diseased mouse model was observed. ICP-MS detected Pd in several organs, including the kidney, liver, and brain for both mouse types, where IMC analysis confirmed these findings in one of the two kidneys tested.

To answer biological questions regarding the effects of NPs on organisms, experiments with higher mouse numbers for better statistics must be conducted in the future. The main purpose of our study was, however, to test and directly compare different, yet complementary analytical methods and show their suitability in the context of Pd-NP detection.

Here, one can conclude that the sensitivity of ICP-MS and IMC are superior to XFI, but the rather complex sample preparation required for the latter two methods might lead to material loss or missing key regions of interest. A combination of all methods by first performing non-invasive XFI measurements, followed by ICP-MS for quantitative results with higher sensitivity, and then using IMC for information down to the cellular level could be a promising workflow for future studies.

Although the Pd-NPs concentrations used throughout this work, especially for the acute exposure experiment, were high and not in a physiologically relevant range, the data provides more detailed biological insights into the uptake and excretion kinetics, as well as accumulation of Pd-NPs in healthy and diseased mice, than has been shown before. We showed that the Pd-NPs applied moved relatively quickly through the GI tract with first particles being excreted as early as 2 h post-gavage. In healthy mice, the highest particle excretion was relatively fast (3–5 h after particle administration). In contrast, in the disease model (APCmin+ mice), the excretion was much slower with high excretion levels at 9 h post-gavage. This could indicate that individuals with GI diseases might not only suffer from a defective intestinal barrier, but also have longer transition times for particles. Taken together, this could result in an increased uptake of particles through the GI tract. However, with subchronic exposure, no statistically significant differences in organ or total Pd recovery were observed between wildtype and APCmin+ mice in our study. Feces collected from healthy mice over 24 h post-gavage showed an almost complete excretion of the initially applied dose. However, in the subacute and subchronic exposure experiments, Pd-NPs could also be found in various organs (especially in the liver and kidney), which indicates that continuous oral Pd-NP intake could lead to tissue accumulation of these particles, with currently unknown consequences. Thus, there is an urgent need to further study the differential uptake-, accumulation-, and excretion patterns of various MNPs to shed light onto this complex and important new field of study. The analytical methods presented here–XFI, ICP-MS, and IMC–alongside model metal-doped Pd-NPs, proved to be a positive advancement in the search for analytical approaches to study plastic particles (even on the nanoscale) in various complex matrices and investigate the spatial distribution of particles down to the cellular level.

## Methods
### Model nanoplastics synthesis and characterization
Pd-doped polystyrene NPs were synthesized in-house as previously described[51]. In brief, NPs were produced via emulsion polymerization, where a Pd-containing salt was introduced together with an initiator into a reactor containing acrylonitrile, SDS, and a polymerizing agent to form NPs with a metal content of approximately 0.3 wt.% Pd. Subsequently, a shell of polystyrene was grown on top of this core particle to achieve a final diameter of approximately 200 nm. Pd-doped NPs were characterized prior to use in terms of particle size, surface charge, and metal content. Particle hydrodynamic diameter and zeta potential were determined by dynamic light scattering and electrophoretic mobility measurements, respectively (Malvern Zetasizer). Full details of particle characterization can be found in Table S13. Previous studies have tested the stability of the Pd-tracer incorporated into the Pd-NPs to ensure no leaching of metal occurred over the duration of experiments, including in biological exposure systems[15,25].

### Animal experiments
All animal experiments were conducted according to Austrian animal welfare legislation, and experimental setups were reviewed and approved by the Federal Ministry Republic of Austria for Education, Science, and Research (BMWFW license 2022.0.257.045). We have complied with all relevant ethical regulations for animal use. Twenty-three male C57Bl/6J (wildtype) and three C57BL/6J-$Apc^{Min}$/J(APCmin+) mice were used at the age of 8–12 weeks (experiment start). Mice were bred in-house and kept under standard conditions (ambient temperature at 12 h light/dark cycle, food, and water provided ad libitium). To investigate the biodistribution of Pd-NPs over different dosages and time points (Fig. 1 and Table S14), three

administration set-ups were applied, including: acute (single dose, 240 min maximum exposure time), subacute (comparison of healthy and colitis-induced mice, 10 days exposure time), and subchronic (subchronic exposure, 5 weeks exposure time). For the acute and subacute experiment, mice from different age-matched litters were randomly assigned to treatment groups at the age of weaning (3 weeks). Litters were equally distributed across treatment groups. For the subchronic exposure experiment, mice were grouped according to their genetic background. Particle solutions for gavage were prepared at respective experimental dilutions in one batch for the whole experiment and aliquots stored at 4 °C. Gavage solutions were briefly sonicated before application in mice to ensure particle dispersion.

For investigation of Pd-NPs acute uptake and biodistribution, ten C57Bl/6J male mice received a single gavage of 100 μL Pd-NPs (3.3817 mg NPs; 9.976 μg Pd). Mice were sacrificed and processed at 5 time points post-gavage: 10, 30, 60, 120, 240 min ($n = 2$ per time point). Urine and stool were sampled for mice with the longest incubation period (i.e., 240 min). To facilitate urine and feces sampling, the mice were put into a cage with a metal grid floor directly after the gavage. Urine was collected on cellulose filter paper. Postmortem, gastrointestinal organs (stomach, small intestine, cecum, and colon) from all mice were tied with twine and processed as described in the respective section below before for shipping for further analysis.

The subacute exposure experiments were conducted in a healthy as well as a colitis-induced model for 10 days. Colitis was induced by applying 2.5% DSS (MP Biomedicals) in sterilized tap water over a five-day period[52], followed by a five-day recovery period with regular sterilized water. Pd-NPs-treated mice received daily oral gavage with Pd-NPs (0.99 mg NPs/day; 2.91 μg Pd/day) for 10 days. Twelve male C57Bl/6J mice were randomly assigned to four treatment groups ($n = 3$), including:
- Group 1 = untreated control (No DSS, No Pd-NPs),
- Group 2 = Pd-NPs (No DSS, Pd-NPs),
- Group 3 = DSS (DSS, No Pd-NPs)
- Group 4 = Pd-NPs+DSS (DSS, Pd-NPs).

At four time points during the experiment, all mice from Group 1 (untreated control) and Group 2 (no DSS, Pd-NPs) were put into cages with a metal grid floor and stool was sampled for 24 h following each gavage (i.e., after 1st/3rd/5th/9th gavage). All mice were sacrificed 24 h after the last gavage (i.e., at day 10), after which the organs were prepared and shipped.

For the subchronic exposure experiments, three male C57Bl/6J and three APCmin+ mice were gavaged for 5 weeks (week 1–4 for 5 days, Monday–Friday with no treatment on the weekend; week 5 gavage only for 4 days, with sampling on the last day, 24 h after the last gavage) with 0.48 mg NPs (~ 1.41 μg Pd/day, 100 μL solution). Feces and urine were collected for 24 h after the last gavage dose by placing mice on a metal grid floor. Mice were sacrificed, then organs were prepared and shipped to collaborating laboratories for further analysis (see below).

Further details on animal experiments and respective reporting according to the ARRIVE guidelines (essential + recommended set of items), which are not mentioned in the main manuscript can be found in the Supplementary Information in Table S15.

## Preparation of mice organs and excretions for uptake measurements

Mice were sacrificed by terminal heart puncture under anesthesia (Ketamin/Xylazin) and blood was collected in ethylenediaminetetraacetic acid (EDTA) tubes (MiniCollect® K3EDTA; Greiner). Organs (lung, heart, liver, kidney, spleen, brain, testis, pancreas) were cut in half and organ weights (total and partial) were recorded. The organ was processed differently for ICP-MS and XFI/IMC measurements. To ensure comprehensive analysis using all three techniques, half of the organ was dedicated to each sample preparation approach. For ICP-MS measurements, the designated tissue was dried on cellulose filter paper at 56 °C for 24–48 h. For XFI and IMC, the tissue was processed according to a modified isopropanol protocol for histological embedding[50]. Due to their small size, mesentery was not cut in

half but processed as a whole and either analyzed with ICP-MS or XFI/IMC. Unless otherwise stated, small intestine and colon were gently flushed with phosphate-buffered saline (PBS) before ICP-MS analysis. For XFI/IMC analysis, small intestine, and colon were additionally flushed with 4.5% paraformaldehyd solution and placed as swiss rolls on filter paper into histocassettes. Small intestines and colons were embedded in a paraffin block, while the other organs were left with a thin layer of paraffin and put into separate tubes for XFI analysis. For the subacute exposure experiment, the non-paraffin organ half was digested for 3 days in 10% KOH at 65 °C with intermediate shaking.

## Quantification and analysis of model Pd-NPs by ICP-MS

The Pd-doping was used as a proxy for the NPs concentration, where Pd was quantified by ICP-MS after microwave acid digestion. As entire samples are digested, there is no possibility for spatial resolution within the tissue, but trace concentrations of analyte (in this case, Pd) can be quantitatively measured. Tissue samples were set on a cellulose filter, and their wet weight was recorded after subtracting the weight of the filter. Subsequently, the tissue was dried and placed in Teflon tubes together with either 3 mL of $HNO_3$ (65% w/w) for smaller organs or 4 mL $HNO_3$ for larger organs. For liquid samples (i.e., blood), the volume of $HNO_3$ was adjusted to achieve 50% w/w $HNO_3$ concentration in the Teflon tubes. Samples underwent microwave acid digestion in an ultraWAVE digestor (Milestone Srl, Italy). The pressure and temperature were ramped from ambient to 220 °C and 160 bar over 30 min and were then maintained for an additional 30 min before returning to ambient conditions. Samples were then diluted by a factor 13:3 in ultrapure water prior to quantification by ICP-MS to reduce total acid concentration. An Agilent 7900 was used for ICP-MS analysis and samples were directly injected into the spray chamber and nebulizer (Glass expansion, Sea Spray 0.4 mL/min) with a peristaltic pump. Measurements were made without the use of a collision cell. The flow rate of plasma gas was 15.00 L/min, the flow rate of the carrier gas was 1.02 L/min and the integration time was 0.5 s.

After standard tuning, the instrument was calibrated daily with 0, 0.1, 0.5, 1.0, 2.5, 5.0, 12.5, and 25.0 μg Pd/L solutions (Sigma-Aldrich, 9975 ± 20 mg/L). Internal standards of yttrium and scandium (10 μg/L) were added directly to calibration solutions and samples prior to analysis and were continuously monitored throughout the measurements to correct for instrumental drift. Pd isotopes 105, 106, and 108 were measured for each sample, but as they all yielded the same values, only [106]Pd is reported hereafter. Final burdens of Pd-NPs in each mouse organ are presented as Pd content (μg) per sample. In the cases where organs were split in two parts, the mass of Pd measured by ICP-MS corresponds only to the mass in the half organ. The ICP-MS LOD and LOQ varied between 14 ng Pd/L and 43 ng Pd/L, 19 ng Pd/L and 56 ng Pd/L, respectively, on different days of analysis (full details see Table S16). Values between the LOQ and LOD were set to the LOD value and values below LOD are indicated as below detection limit (bdl). Sample LOD/LOQ values [ng] were calculated using the instrumental LOD/LOQ values [μg/L] multiplied by the total digested sample volume [mL], which was depending also on the distinct dilution/preparation for the various sample types. For the subchronic exposure experiment, assessed Pd-NP organ concentrations were compared between wildtype and APCmin+ treatment groups using an unpaired Student's t-test (software GraphPad Prism version 8.0.1).

## Experimental setup, data acquisition, and analysis of synchrotron-based XFI measurements

All XFI measurements were executed at the P21.1 beamline of the PETRA III synchrotron (DESY, Hamburg, Germany) utilizing a monochromatic incident X-ray energy of approximately 53 keV. This energy was a good compromise between background reduction and fluorescence cross-section[40]. The samples, prepared as described above, were mounted on a dedicated holder to perform a 2D scan transversal to the incident X-ray beam of 1 mm × 1 mm area, resulting in a spatial resolution of 1 mm. A total of 10 silicon drift detectors (SDDs) of 50 mm² collimated area and 1 mm

sensor thickness from Amptek (XR-100 FAST SDD, Amptek Inc, MA, USA) were positioned around the target at a distance of 6 cm from its center and under detection angles ranging from 30° up to 150° in 30° step sizes. An image of the experimental setup, which consisted of a custom-made imaging platform with a translation arm to allow scanning of the samples, is shown in Fig. S5. The platform was positioned on a diffractometer provided by the beamline in the experimental hutch and aligned precisely with respect to the incident X-ray beam. For accurate alignment of the samples, an on-axis camera is used of which the mirror with a hole in the center indicates the direction of the incident beam.

Since synchrotron beamlines typically do not provide absolute photon flux numbers, several different dedicated reference targets from Micro-matter (Surrey, BC, Canada) were used for calibration purposes. Since the layers deposited on the targets have well-defined masses, the measurement of emitted fluorescence photons allowed for a reconstruction of the incident photon flux, which was essential to quantify the Pd in each sample. Given the fact that the incident energy and photon flux strongly depend on the beamline configuration, it can vary between measurement campaigns, as shown in Table S17, which summarizes the main experimental parameters.

To accurately determine the number of Pd photons, peak fitting was performed[53]. The number of reconstructed fluorescence photons, in combination with the incident photon flux, allowed for the determination of Pd masses contained within the samples. However, it must be noted that because all samples were scanned in a 2D mode, no attenuation within the samples was considered here. A more precise mass calculation would require detailed knowledge about the sample geometry, especially its density distribution in 3D, which was not acquired in this case. Therefore, all reconstructed masses underestimate the actual Pd values by approximately 10–25% which was determined by calculating the attenuation for Pd fluorescence in representative geometries.

To validate the administered Pd masses in each gavage, representative aliquots of the injected solution were quantitatively analyzed for some experiments, revealing a strong correlation between reconstructed and anticipated values from ICP-MS measurements.

## Imaging mass cytometry (IMC)—data acquisition, quality control, and sensitivity analysis

In IMC, a tissue slide is inserted into the Hyperion™ Tissue Imager (HTI, Standard BioTools) where a 213–219 nm UV laser ablates the tissue at a frequency of 200 Hz in consecutive 1 μm diameter shots (Fig. S6). The released material is carried on a plume of helium and argon gases into the mass cytometer (MC) for detection in the time-of-flight (TOF) chamber. The MC is an inductively coupled plasma TOF mass spectrometer tuned to detect heavy metals in the mass range of 76–209 (135 possible channels). Tissue images are reconstructed at 1 pixel per 1 μm laser shot and saved as mass cytometry data (MCD) and TXT files. The NPs in this study are tagged with palladium containing the naturally occurring distribution of isotopes ($^{102}$Pd, $^{104}$Pd, $^{105}$Pd, $^{106}$Pd, $^{108}$Pd, $^{110}$Pd).

To detect Pd-NPs by IMC, organs were prepared as follows: formalin-fixed paraffin-embedded (FFPE) tissues with 3.5 μm thickness were dewaxed as follows: Xylol $2 \times 10$ min, 100% EtOH $2 \times 5$ min, 94% EtOH $1 \times 5$ min, 70% EtOH $1 \times 5$ min, then in deionized water (dH$_2$O) for 5 min on a shaker with gentle agitation. Slides were rinsed in PBS on a shaker with slow agitation for 10 min and then stained with iridium ($^{193}$Ir) DNA intercalator at 1:200 (cat. 201192B, Standard BioTools) for 30 min at room temperature in a hydration chamber. Slides were washed 3 times in dH$_2$O and air-dried for at least 20 min at room temperature under air flow in a hood before acquisition on the Hyperion.

IMC acquisition begins by connecting the argon and helium gas tubing, starting up the machine and software, and igniting the plasma. Quality control (QC) was performed by running an automated QC program on a 3-element coated tuning slide ($^{89}$Y, $^{140}$Ce, $^{175}$Lu) (cat. 201088, Standard BioTools). Machine settings (e.g., detector voltage, gas concentrations, cone voltage) were calibrated and optimized and then the sample slide was loaded into the Hyperion. A new MCD file was created, and a slide image (taken with a mobile phone) was aligned with the camera view image in the CyTOF software (version 7.0.8493) so that the laser position indicates the same region in both images. A panorama high-resolution image was then made to support ROI discrimination, and the ROIs are selected, the panel template created, and the samples run. In the acute exposure experiment, nine ROIs were acquired from the brain, four from the liver, and eleven from the small intestine. In the subacute exposure experiment, one ROI from a single mouse from each condition (untreated control, Pd-NPs, Pd-NPs+DSS) was acquired from the colon, small intestine, liver, and kidney. In the subchronic exposure experiment, mouse #29199 had four ROI from the brain and three from the kidney and mouse #29207 had three ROI from the kidney. All ROI were ablated at 200 Hz laser frequency and 3 decibels (db) laser power with a scan area timing of $\geq 1$ mm$^2$/2 h. The areas of the ROIs ranged from 0.5 to 1 mm$^2$ and all Pd mass windows, as well as iridium (for DNA), were opened for measurement.

To check data quality, the MCD raw data file was opened in the napari image viewer with the napari-IMC plugin. First, the iridium signal was displayed in blue for visualization of the cell nuclei and all the Pd channels in yellow, which indicates the presence of NPs. We confirmed that the nuclei were well resolved and that the Pd signals were amplified as much as possible to achieve a lower detection limit. After, the data was loaded into R Studio and a spatial experiment was created using the IMC data analysis workflow R script. We used the imcRtools package[54] and displayed DNA in blue and the 5 highest expressed Pd isotopes in yellow. The number of Pd pixels per ROI (termed dual counts on the $y$-axes) for $^{108}$Pd was created with the Plotpixels package.

In mass cytometry, the output of the detector is in dual counts. This is a dual scale encompassing single ion counts for low-level signals and intensity for higher signals. For the purposes of this study, the dual counts detected were low enough to reflect single particles, called pulses, hitting the detector. That is, if one has 10 dual counts of a given metal, this equates to 10 ions of that metal detected.

To identify the LOD of Pd-NPs on the Hyperion tissue imager, a titration experiment was performed with 91 ng Pd/μL suspension of Pd-NPs (Figs. S7 and S8). First, the slide was coated with 2% agarose, as described[55]. Then 1 μL of Trypan blue was added to 10 μL NPs suspension and a 1:1 dilution series was performed 40 times. Each dilution was spotted onto a slide at 0.2 μL per spot. The concentration of Pd was 18.2 ng in spot 1, and this was diluted to $1.09 \times 10^{12}$ in spot 40. The spots were acquired on the HTI in $10 \times 200$ μm ROIs and Pd channels 110, 108, 106, 105, 104, and 102 were open to detect signals. Data was first visualized in Napari and then analyzed in R with an edited version of the spill correction script from the IMC data analysis workflow (Fig. S8)[54].

In the heatmap plot from the titration experiment, the dual count signals from $^{108}$Pd and $^{106}$Pd were the strongest, indicating they were the most abundant isotopes in the Pd-NPs (Fig. S7). The least diluted samples had extremely high Pd content and consequently contained many Pd-NPs per pixel. Conversely, the most diluted samples likely contained single Pd-NPs because the likelihood of aggregation is low, and the samples were extremely diluted (Fig. S8). A range of 1- 10 dual counts of Pd was seen in the pixel counts graph in Fig. S8C, with most of the signal just below 5 counts. This indicates that, on average, one ion of Pd correlates to 1 dual count and up to 4 Pd-NPs per pixel can be reached at high concentrations. Though metal loading across all individual Pd-NPs was very similar[16], even slight differences could lead to variations in IMC measurements. Supporting this, on the low end of the titration (40×), when single NPs might be in the sample, signals at approximately 2 pixels were detected.

## Reporting summary

Further information on research design is available in the Nature Portfolio Reporting Summary linked to this article.

## Data availability

Source data can be obtained in the Supplementary Information and from the following Zenodo repository, https://doi.org/10.5281/zenodo.16629231. Other data are available from the corresponding authors upon request.

## Code availability
All scripts and custom code are available upon request to the corresponding authors.

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

## Acknowledgements

L.K., V.P., and V.K. acknowledge the support from microONE, a COMET Modul under the lead of CBmed GmbH, which is funded by the federal ministries BMK and BMDW, the provinces of Styria and Vienna, and managed by the Austrian Research Promotion Agency (FFG) within the COMET—Competence Centers for Excellent Technologies—program. A.P. was funded by the ETH Postdoctoral Fellowship and the Rütli Foundation. DMM was funded through the Swiss National Science Foundation (grant number PCEFP2_186856). We would like to thank Dana Leuenberger from the Visceral Surgery Research Lab for her help processing the IMC samples and the IMC titration experiment as well as the IMC Platform in Bern for their acquisition of the IMC samples. We are grateful to Tural Yarahmadov for his support editing the R script for the IMC analysis as well. We thank Edina Gashi and Thill Wollenmann for their help with the ICP-MS measurements and also thank Marte Haave for fruitful discussions on various study designs. Parts of this research were carried out at PETRA III and we would like to thank the beamline team for their assistance in using the P21.1 beamline. We thank Markus Schink for validating our graphics color scheme for readers with color blindness. We acknowledge financial support from the Open Access Publication Fund of Universität Hamburg.

## Author contributions

T.S. (conceived experiment, performed experiment and data analysis, manuscript writing), V.K. (conceived experiment, performed animal experiments, data analysis, manuscript writing and editing), A.P. (performed experiment and data analysis, manuscript preparation), T.B. (conceived experiment, sample acquisition, data analysis, manuscript preparation), R.K. (data analysis, preparation and compilation of figures, manuscript preparation), D.S. (conceived experiment, provided laboratory and infrastructure support), J.W. (performed animal experiments), L.K. (study design, resources, supervision, manuscript preparation and editing, funding acquisition, animal study design and approvals), V.P. (conceptualization, data analysis, manuscript and figure preparation and editing), F.G. (conceived and performed experiment, acquired funding, supervision), D.M.M. (conceived study design, designed analysis, data analysis, manuscript writing, acquired funding, supervision).

## Funding

## Competing interests

The authors declare no competing interests.
