## [Transparent Peer Review file · Communications Biology]

Biodistribution of Nanoplastics in Mice: Advancing Analytical Techniques Using Metal-doped Plastics

Corresponding Author: Dr Theresa Stauffer

Version 0:

Reviewer comments:

Reviewer #1

(Remarks to the Author)

This study, by Dr. Stauffer et al, is centered around the biodistribution of nanoplastics in mice and provided insights into the kinetics of nanoplastics distribution in an organism at different exposure scenarios. However, I have some comments, which should be addressed.

1. Regarding the sensitivity issue that the authors pointed out, please compare how it differs from existing analytical methods.
2. When quantifying metals in organs, is perfusion performed as pretreatment? Is it possible that the author measured the amount of metal in the blood that remains in the organs?
3. When discussing particle uptake and distribution, results should be presented showing changes in the amount in the blood over time. If particles migrate into the blood and are then excreted in the bile or urine, it seems that the particles would be detected not only in the liver and kidneys but also in the blood. When considering rapid distribution to organs, I think it is essential to discuss kinetics such as the rate of disappearance from the blood and half-life. The author mentioned that the particles migrate from the digestive tract to the feces, but if they have once migrated into the digestive tract, they should be detected in the blood and liver. It is therefore difficult to discuss particle dynamics based only on time-dependent changes in organs.
4. If metals are trapped in the small intestine and do not reach the blood or liver, by what route do you think particles that have been taken into the small intestine are excreted in the feces?

Reviewer #2

(Remarks to the Author)

The paper "Biodistribution of Nanoplastics in Mice: Advancing Analytical Techniques Using Metal-doped Plastics" written by Stauffer et al. shows biodistribution of nanoplastics (NPs) in mice with using Pd-doped NPs. The results were well presented, and the research topic is attractive. However, the following points need to be improved.

1. Differences between metal-doped NPs and non-metal-doped NPs

The authors accurately describe Pd-NPs distinguishing them from normal NPs throughout the manuscript, and I understand that this study is based solely on the results of using Pd-NPs. My question is whether the ADMET of metal-doped NPs is the same as that of non-metal-doped NPs. Table S7 lists information on Pd-NPs, but I think my question would be answered if there was information on non-Pd-NPs for comparison. In addition to the information in Table S7, it would be good to have information on the density of Pd-NPs.

In addition, if the authors' thoughts on the homology or heterogeneity of the physicochemical properties of metal-doped NPs and normal NPs were presented, and if they were heterogeneous, a discussion on the impact on ADMET would help expand the use of metal-doped NPs.

2. The necessity of animal experiments using disease model mice

In the introduction, there is no explanation as to why model mice were not used in the short-term study and why different model mice were used in the intermediate-term and long-term experiments. In addition, in the intermediate-term study, no results were shown that could compare the differences between disease model mice and normal mice (Fig. 4), which seems to diminish the significance of the study using disease model mice. In general, this is a paper about a method using Pd-

doped NPs, and if the authors want to promote this point, I think they need to clarify my first comment. Also, if they want to promote the results using model mice, I think it would be better to show figures and tables that make it easier to compare.

2. No statistical tests were performed.

In animal experiments, it is common to require appropriate statistical analysis from the standpoint of scientific validity and ethics. This is a critical problem of this study.

3. The necessity of single particle analysis

Although there is no leach of Pd from Pd-NP (Line 467-469), it seems that Pd-NP can be detected more directly by performing single particle ICP-MS measurement. Depending on the particle size of the core Pd, it may be possible to detect lower concentrations of Pd by using the single particle ICP-MS method. Therefore, I think it would be a good idea to include single particle ICP-MS measurement in the future.

4. Term of "intermediate" and "long"

Generally, the duration of toxicity tests to evaluate long-term toxicity is 6 months or more. Also, the duration of toxicity tests to evaluate subacute toxicity is 1 to 3 months. Therefore, I think it would be a good idea to revise the names of the administration periods according to the general definition of animal experiments.

5. The experimental term and the displayed term are different.

The intermediate term is a 10-day administration period (Fig. 1), but Fig. 4b only displays up to 9 days. Also, the long term is 5 weeks (Fig. 1), but Fig. 5b only displays up to 10 days.

6. Regarding the results of dual counts by IMC (Fig. 3d, 4d, 5e)

It is difficult to distinguish the points of the dot plots shown, and it is unclear in which conditions/organs Pd is accumulating. It would be a good idea to modify the data so that the average and distribution of the data can be visualized by making the Y-axis logarithmic, adding jitter settings to the dot plot, combining the dot plot with a box plot/violin plot, etc.

Also, does this mean that the results show the Pd signal intensity in all pixels in the ROI, and that the data for pixels where Pd is not accumulating is also included? It would be better to clarify whether the results show only data above the LOD.

7. "dual counts"

I think the term "dual counts" refers to the pulsed mode and analog modes of the detector of ICP-MS, but normally the modes are switched automatically, and signal intensities are provided as count (cps) in both modes. I think it would be sufficient to describe how the authors adjusted for the difference in sensitivity between the pulsed mode and analog mode and write "signal intensity (count)".

Or does Cy-TOF always output both modes signals? If so, it would be better to explain how you processed the signal intensity. For example, the average value of the pulsed and analog modes was used. Or if the pulsed-mode signal exceeded 100 counts, for example, the analog-mode signal was used.

Minor comments

8. Fig. 3a

Wouldn't it be better if the optical image was also shown?

9. Regarding the IMC tissue images (Fig. 3c, 4c, 5d)

Please provide the units for the scale bars in these figures. Also, please include the color bars and their units. The yellow pixels are difficult to see. Please enlarge them or add arrows so that we can see the specific areas.

10. Fig. 4d

Please specify whether the control shown here is an untreated control or a colitis control.

11. Fig. 5e Insufficient information on the x-axis

For example, Brain1 shows four dot plots. Please explain what is different.

12. Mass numbers are generally written in the left superscripts

13. Regarding the measurement of Pd using ICP-MS

It is said that similar values were obtained by monitoring $m/z = 105$, 106 , and 108 (Line 542), but it is unclear which results were used as the basis for the discussion.

Also, since there is spectral interference on $m/z = 106$ and 108 from 106Cd and 108Cd , it seems appropriate to use the results for $m/z = 105$. Also, was the measurement performed in KED mode? It would be better to clarify the measurement conditions, such as the flow rate of the CRC gas.

Reviewer #3

(Remarks to the Author)

This study proposes a workflow that combines three methods (ICP-MS, XFI, IMC) with using nano-PS particles labeling with Pd to measure the residual NP in test animals. The experimental design using NP inside-including Pd for minimizing influence of labeling on the residual behavior in the body and using three different analytical methods is interesting. However, for the following reasons, the novelty and significance of this proposal for the investigation of the residual behavior of NP have not been sufficiently demonstrated.

Major points

1. A clear comparison of the analytical merits of the three methods (ICP-MS, XFI, and IMC) employed in this study should be made for understanding the proposed workflow and selection of order of the methods: 1) between the three methods. For example, detection limit as numbers of NPs as well as Pd content, spatial resolution, and volume or weight concentration, quantification and its reproducibility, and measurement interferences. Comparisons under identical sample conditions are useful if possible. 2) Performance comparison between these three methods and previous methods (e.g., IR, pyrolysis GC/MS, microscopic observation of particles, etc.) 3) comparison of time, and complexity of procedure, cost of the analytical methods.

2. In the workflow scheme, XFI is proposed as a primary screening method for detecting NP-Pd, however, the application range is not clearly stated. It has a lower detection sensitivity than IMC and ICP-MS, and there is also a possibility that NP-Pd particles will be lost due to pretreatment and cleaning, so it may be unsuitable for primary screening because there is a possibility that residual NP-Pds in rat will be overlooked.

3. The NP-Pd particles in the particle solution that are ingested by the rats need to be dispersed and homogenized for the purpose of this study, but this information is not available and cannot be verified. The residual behavior may change depending on whether the NP-Pd particles are dispersed or aggregated.

4. Is the homogeneity of the samples of rat organs and feces after exposure guaranteed between the three methods (ICP-MS, XFI, IMC)? The manuscript mentioned that there is a possibility of loss of NP-Pd particles due to washing operations, etc., but have it been verified the exclusion ratio into the washing solution? Also, the manuscript mentions that the organs were divided into two parts, but how were they divided? If it is the upper and lower parts of the organ, is there a possibility that it will accumulate in one part depending on the type of organ?

5. The reason for setting the amounts of exposure doses for rats is not clearly stated. For example, the dose for the short-term study seems to be high, but is it exceeding the normal range of rat physiology and causing abnormal excretion behavior?

6. The superiority of the residual measurement method using PdNP in this study is unclear. It should be compared the behavior of other types of NPs (e.g., only PS, NPs with another tag molecules, metal nanoparticles) in previous studies.

Minor points

7. Rat experiments:

Did the rats consume the entire amount of the garbage solution? In the high-dose condition, did the rats not immediately vomit due to a sense of foreign matter? Also, are multiple rats kept in one test cage? If so, is there a possibility of cross-contamination due to accidental ingestion of feces from other rats ?

8. Condition of ICP-MS instruments at the ICP-MS and IMC methods should be added:

the conditions of ICP-MS instruments such as flow rates of plasma and carrier gas, and integration time. the condition for IMC such as wavelength and pulse speed of ablation laser, sample scanning speed.

9. L576-587, XFI measurement:

Is the depth that can be measured equivalent to the thickness of the sample? If it is only the surface, is the sample's inhomogeneity not affected? Also, to check the degree of a strong correlation between reconstructed and anticipated values from ICP-MS measurements, please show specific data.

10. Table S7:

Since it is a group of particles, the hydrodynamic adiameter should show the distribution, not just the average value.

Version 1:

Reviewer comments:

Reviewer #1

(Remarks to the Author)

I think the expression "in the digestive tract" is the cause of the misunderstanding. Does this mean that the particles penetrate the intestinal barrier and were detected in the tissue, but rather that they were detected in a mixture of contents and tissue? I was unable to determine whether it was in the lumen or tissue.

If the particles passed through the intestinal barrier, it may be in the blood. Furthermore, even if the particles were not detected in the blood at a certain time point, it cannot be denied that this is the result of rapid distribution to organs. Therefore, I thought that tracking the kinetics in the blood over time and calculating the elimination half-life, etc., would be

useful in solving this question. Moreover, if the particles that passed through the intestinal barrier are excreted in the feces, it is possible that they were transferred to the feces via bile excretion from the liver.

Reviewer #2

(Remarks to the Author)

The authors adequately responded to the reviewers' questions. Most of my questions have been resolved. However, I think the following corrections are necessary.

1. Model animal

The authors used an animal model of small intestinal disease. I want the authors to explain in the introduction why they focused on the small intestine and the relationship between the disease and NP.

2. Fig. 5e

"ROI 4 – 7" should be "ROI 5 - 7"

Reviewer #3

(Remarks to the Author)

I would like to appreciate for your response and revision to the reviewer's comments. However, a few questions regarding some parts of your response are remained. I would like to recomend to clarify the following points and revision of the manuscript.

Comments to the authors' answers

To Answer 1.

At the columns about reproducibility in Table S7, specific examples of numerical data should be added to clarify the comparison.

To Answer 2

Regarding XFI, the answer states that XFI is non-destructive and does not require special sample treatment. While the embedding treatment is correct for tissue preservation, there is no information provided to determine whether the preservation of nanoparticles remained at organs is acceptable, making it impossible to judge. The embedding treatment used isopropanol and additionally paraformaldehyd solution for the small intestine and colon. At the very least, the organs have been in contact with these solutions, and the possibility of particle loss cannot be excluded. As requested in the original comments from the reviewers, the effects of particle loss due to contact with these solutions should be discussed and the revision of the manuscript is recommended. Furthermore, although the limitations due to lower detection sensitivity of XFI compared to other methods are mentioned in the answer, this should also be clearly stated in the manuscript. In addition, Table S7 has been added, so it is recomend that the advantages and disadvantages of XFI, ICP-MS, and IMC are discussed at each item (e.g., time, cost, and reproducibility).

To Answer3

Although an operation to promote particle dispersion has been ceratnly added, it has not been confirmed whether the particles were dispersed. I recommend an evidence of the dispersion is added for the confirmation.

To Answer4

It claims that the agreement between the XFI and ICP-MS measurements indicates that particle loss due to washing is minimal. However, if the influence of leaching due to pretreatments in both XFI and ICP-MS is significant, and even under conditions where the extent of influence is comparable, the agreement between XFI and ICP-MS analysis results may still hold. Therefore, returning to the original reviewer's comment, it is necessary to evaluate or estimate the particle loss effect of each pretreatment for XFI or ICP-MS under the experimental condition.

To Answer 6

Thank you for adding examples of conventional labeled NPs. However, please cite each reference for labeling with fluorescent dyes, inorganic metals, radioactive substances, or stable isotopes of carbon polymers for understanding.

To Answer 8

Information on laser ablation has been added, but important laser wavelengths are missing. In addition, laser output should be normalized per area and expressed in units such as J/cm².

Version 2:

Reviewer comments:

Reviewer #1

(Remarks to the Author)

The authors have responded appropriately to the reviewers' comments and the questions have been resolved. Thank you for your consideration.

Reviewer #2

(Remarks to the Author)

The authors have responded comprehensively to the reviewers' suggestions, and the revised manuscript represents a valuable contribution to the field. I recommend that the paper be accepted for publication.

Reviewer #3

(Remarks to the Author)

The authors responded adequately to the reviewers' comments and revise the manuscript accordingly.

Reviewer #1 (Remarks to the Author):

This study, by Dr. Stauffer et al, is centered around the biodistribution of nanoplastics in mice and provided insights into the kinetics of nanoplastics distribution in an organism at different exposure scenarios. However, I have some comments, which should be addressed.

1. Regarding the sensitivity issue that the authors pointed out, please compare how it differs from existing analytical methods.

Answer: *As elaborated in the Introduction (lines 61 onwards), several analytical challenges exist when it comes to the assessment of nanoplastics (NPs) in biological samples. To overcome the issues described, we suggest using the combination of XFI, ICP-MS and IMC to quantify metal-doped model NPs. While not directly suitable for assessing environmental NPs, this approach is well suited for laboratory studies to understand relevant mechanisms and processes. A comparison of the main parameters of all three methods has now been added to the Supplementary Information (Table S7).*

Table S7: Comparison of the analytical merits of the three methods employed in this study

Method	Limit of detection (LOD) & quantification (LOQ)	Spatial resolution	Measurement interferences	Reproducibility	Time investment	Access to equipment
XFI	2.2 - 36.6 ng/sample Full list see Tables S1, S3 and S5, as the detection limit strongly depends on the object of investigation	Defined by beam diameter; here 1 mm but can be much lower (nm-scale at dedicated synchrotron beamlines)	None	High	18 min for full-body scan of a mouse in vivo; scan time can vary depending on the required spatial resolution and size of the object of investigation.	Synchrotron for high-sensitivity measurements is needed but compact systems are under development. Labeling of particles but no other sample preparation needed
ICP-MS	LOD: 0.014 - 0.019 µg/L LOQ: 0.043 - 0.056 µg/L (instrumental) LOD: 0.14 - 0.30 µg/L LOQ: 0.43 - 0.88 µg/L (blood samples) LOD: 0.42 - 1.43 µg/L LOQ: 1.29 - 4.20 µg/L (organ samples) LOD: 0.19 - 0.43 µg/L LOQ: 0.56 - 1.26 µg/L (feces samples) Detailed list see Table S10	None: Spatial resolution is generated by dissecting and separating different organs	None (but depending on elemental distribution of the sample, spectral interferences may be possible in ICP-MS in some cases but often can be corrected for)	Measurements of digested samples are highly reproducible. However, reproducibility can vary due to the sample preparation requirements (e.g.: in this case issues recovering stool)	Microwave acid digesters vary in sample capacity, and consequently the time required to measure a given number of samples varies. After instrument daily checks, ICP-MS measurements take approximately 2 minutes per sample. With our current laboratory infrastructure, from digestion to analysis, approximately 30 samples can be measured in a day.	Both microwave acid digestion and ICP-MS analysis are common and routine analytical devices and procedures for laboratories suited to trace metal analysis
IMC	In the context of proteins detected on single cells with antibodies tagged to a metal, the LOD is ≥ 400 copies/ μm^2 , however, based on the titration experiment in this study, one Pd-NP is estimated to be one dual count per pixel	Pixels are displayed with their x and y position in space on a tissue slide, so the position of every pixel is known and is $1\mu\text{m}^2$	None	High	2 hours per 1 mm^2 tissue area acquired	There are 3 facilities in Switzerland that offer Hyperion services

2. When quantifying metals in organs, is perfusion performed as pretreatment? Is it possible that the author measured the amount of metal in the blood that remains in the organs?

Answer: *Organ perfusion, by means of PBS and PFA flushing through the animals' blood circulatory system, was not performed. However, terminal heart puncture was performed, thus reducing the "left-over" blood in the circulatory system and the organs. Pd concentrations in the blood were also measured, and values are given in Tables S1, S2C, S3, S4B, S5, S6B and S6C. For most samples, Pd was below the analytical detection limits. Furthermore, Pd is a rare heavy metal not commonly present in most environmental or biological samples, and together with blank controls, we can be confident that the detected Pd-levels reflect the presence of model Pd-doped NPs.*

3. When discussing particle uptake and distribution, results should be presented showing changes in the amount in the blood over time. If particles migrate into the blood and are then excreted in the bile or urine, it seems that the particles would be detected not only in the liver and kidneys but also in the blood. When considering rapid distribution to organs, I think it is essential to discuss kinetics such as the rate of disappearance from the blood and half-life. The author mentioned that the particles migrate from the digestive tract to the feces, but if they have migrated into the digestive tract, they should be detected in the blood and liver. It is therefore difficult to discuss particle dynamics based only on time-dependent changes in organs.

Answer: *We are not sure if we understood the reviewer's comments correctly, but we want to clarify that orally ingested particles do not need to enter the bloodstream or organs before being excreted. Our short-term/acute exposure experiment shows that over 70% of the administered dose remains within the gastrointestinal tract (GI; stomach > small intestine > colon) up to 4 hours post-application, and peak excretion via feces was detected between 4–5 hours after gavage. This suggests that most particles transit the GI tract along with digested food and are excreted, with only minimal uptake through the intestinal wall. Also, previous studies^{1–3} report low intestinal absorption rates (< 0.3%) for particles $\leq 3 \mu\text{m}$. Consistent with this, we detected Pd signals in only 2 out of 16 blood samples, indicating that NP uptake into the bloodstream is minimal and often below our methods' detection limits. Consequently, we cannot reliably assess particle uptake and distribution kinetics in blood over time. Please also note that the total blood volume analyzed was only 100 μL out of a total volume of roughly 1.7 mL per mouse, so it is possible that we have not been able to detect the NPs present in the blood.*

4. If metals are trapped in the small intestine and do not reach the blood or liver, by what route do you think particles that have been taken into the small intestine are excreted in the feces?

Answer: *We may have misunderstood the reviewer's comment, but we want to state that orally ingested particles do not need to enter the bloodstream or organs before being excreted. Similar to ingested food, particles can transit the GI tract (mouth, stomach, small intestine, colon, rectum) and are excreted in the feces. However, depending on size, composition, surface properties and hydrophilicity, particles could interact with the mucosal layer in the gut, be taken up through the GI tract into the bloodstream, and thus transit the body at different rates. Particles taken up into the bloodstream might be removed via liver and bile (excreted into feces), whereas particles that could appear in the lymph (< 0.2 μm) could be eliminated through the splenic filtration system into the gut⁴.*

Renal excretion (kidney/urine) might only be possible for smaller particles < 1 μm due to size limits in the glomerular filtration barrier.

Reviewer #2 (Remarks to the Author):

The paper "Biodistribution of Nanoplastics in Mice: Advancing Analytical Techniques Using Metal-doped Plastics" written by Staufer et al. shows biodistribution of nanoplastics (NPs) in mice with using Pd-doped NPs. The results were well presented, and the research topic is attractive. However, the following points need to be improved.

1. Differences between metal-doped NPs and non-metal-doped NPs

The authors accurately describe Pd-NPs distinguishing them from normal NPs throughout the manuscript, and I understand that this study is based solely on the results of using Pd-NPs. My question is whether the ADMET of metal-doped NPs is the same as that of non-metal-doped NPs. Table S7 lists information on Pd-NPs, but I think my question would be answered if there was information on non-Pd-NPs for comparison. In addition to the information in Table S7, it would be good to have information on the density of Pd-NPs.

Answer: *We acknowledge that micro-and nanoplastics are a large suite of contaminants with different properties including polymer type, surface chemistry, shape, extent of weathering and others. These different physiochemical properties may also lead to variations in uptake and toxicological impacts, but the extent of this variation is not yet known and is currently under investigation by many scientists in the field. In this case we have chosen to use a model nanoplastic with defined properties administered as a homogeneous suspension. These materials are a good proxy for environmental nanoplastics and the simplification to choose one model material was necessary for methodological reasons given the relatively large number of exposures which were conducted throughout the experiment presented in our manuscript. While plastic particles with different physiochemical properties may have different degrees of uptake and impart different effects, we have been mindful to not over-extend the findings of this study and have been careful to draw conclusions based on the model materials we used here.*

The model metal-doped nanoplastics' polystyrene surface composition and their irregular shape are environmentally relevant. Furthermore, the metal-dopant is contained within the core (in polyacrylonitrile) and therefore does not interact with the organism and no Pd leaches from the particles over the exposure. This makes them a satisfactory proxy for real nanoplastics. What makes these Pd-NPs less environmentally representative is the presence of surfactants which are otherwise not present in environmental nanoplastics. However, environmental nanoplastics will also generally be coated with natural organic matter which has surfactant-like properties⁵.

The density of polystyrene is 1.05 g/cm³ and the density of polyacrylonitrile is 1.18 g/cm³. Therefore, the nanoplastics will have an intermediate density between these values. However, please note that since nanoplastics are colloidal particles, by definition, their behavior is not affected by density, which is also the reason why we have not included this information in the main text.

We have added the following paragraph to better explain why we have chosen to use model metal-doped nanoplastics:

Lines 74- 81: When choosing model MNPs to study, one must keep in mind that MNPs are a large suite of contaminants with different properties such as polymer type, surface chemistry, extent of weathering and others. These different physiochemical properties may also lead to variations in uptake and toxicological impacts, but the extent of this variation is not yet known and is currently under investigation by many scientists in the field. Therefore, we have chosen to use model nanoplastics with defined properties administered as a homogeneous suspension. The model metal-doped materials are a good proxy for environmental nanoplastics due to their surface composition and irregular surface.

2. The necessity of animal experiments using disease model mice

In the introduction, there is no explanation as to why model mice were not used in the short-term study and why different model mice were used in the intermediate-term and long-term experiments. In addition, in the intermediate-term study, no results were shown that could compare the differences between disease model mice and normal mice (Fig. 4), which seems to diminish the significance of the study using disease model mice. In general, this is a paper about a method using Pd-doped NPs, and if the authors want to promote this point, I think they need to clarify my first comment. Also, if they want to promote the results using model mice, I think it would be better to show figures and tables that make it easier to compare.

Answer: *The short-term/acute exposure scenario was only conducted in healthy mice as a proof of principle study to get a first idea of particle movement through the intestinal tract, about systemic uptake and excretion and to determine if we can detect signals at all with the analytical methods applied through the rest of the study. For this, we purposefully used higher nanoplastics (NPs) concentrations which could be considered less physiologically relevant. However, more realistic NP exposure is likely to occur on a chronic basis at much lower concentrations, thus we designed the subsequent experiments (where we also included disease models) with more realistic doses and with longer exposure schemes.*

In the Results section in the first paragraph for every experiment, we highlighted the specific rationale behind the experimental design and what we aimed to analyze, or why we chose certain conditions or models in more detail. This includes the use of disease models for longer exposures. To further clarify this based on the reviewer's comment, we additionally reformulated one paragraph in the Introduction to state the use of the disease models clearly up front and added a more detailed discussion on the comparison of wildtype and APCmin+ mice in the Results and the Discussion section.

Lines 184 – 186: This included healthy mouse models of Pd-NPs exposure as well as disease models of colitis for subacute exposure and APCmin+ (intestinal polyps) at subchronic exposure to also cover the influence of certain intestinal diseases on NP uptake and tissue accumulation (see schematic in Figure 1).

Lines 517-525: Although single APCmin+ mice exhibited quite high Pd signals in distinct organs (e.g. brain, liver), the overall Pd levels per organ were comparable to those of wildtype mice, with the exceptions of heart and spleen. In heart samples surprisingly higher Pd concentrations were detected in wildtype mice (wildtype 2.81% vs APCmin+ 0.34% of daily dose, p=0.073), whereas for spleen samples all three APCmin+ mice had higher Pd concentrations (wildtype 0.19% vs APCmin+ 0.87,

p=0.122). However, both findings could not reach statistical significance with the low number of animals used in this study. Also, total Pd recovery rates per mouse were observed to be in a comparable range between wildtype and APCmin+ mice.

Lines 684-686: However, with subchronic exposure no statistically significant differences in organ or total Pd recovery were observed between wildtype and APCmin+ in our study.

2. No statistical tests were performed.

In animal experiments, it is common to require appropriate statistical analysis from the standpoint of scientific validity and ethics. This is a critical problem of this study.

Answer: *The main focus of our study was the direct comparison of promising analytical methods to assess the biodistribution of model nanoplastics in tissues. Therefore, the whole study was designed in a way that animal experiments served to show that the basic principles of all methods work, using only 2-3 animals per group. Also, for some samples, like feces, we only obtained one sample per timepoint, as this was collected from multiple animals. Due to the limited sample number, we decided that statistical tests would not deliver meaningful results, which is why they were not performed a priori in our study. A comparative analysis of nanoplastic uptake and organ distribution among e.g., healthy and diseased mice models with a reliable statistical analysis is beyond the focus of our study and would require at least 5 mice per group and should thus be investigated in a follow-up experiment. However, we appreciate the comment given and performed statistical testing of the ICP-MS data for wildtype and APCmin+ treatment groups in the subchronic exposure experiment. Although we did not observe any statistical differences between the two groups, we also added a more detailed description of the differences between wildtype and APCmin+ mice, together with p-values in the text for the most outstanding differences between the two groups. For the respective paragraph added to the Results and Discussion part, please see our answer to question #1, above. To the Methods we added the following line:*

Lines 826-828: For the subchronic exposure experiment, assessed Pd-NP organ concentrations were compared between wildtype and APCmin+ treatment groups using an unpaired Student's t-test (software GraphPad Prism version 8.0.1).

3. The necessity of single particle analysis

Although there is no leach of Pd from Pd-NP (Line 467-469), it seems that Pd-NP can be detected more directly by performing single particle ICP-MS measurement. Depending on the particle size of the core Pd, it may be possible to detect lower concentrations of Pd by using the single particle ICP-MS method. Therefore, I think it would be a good idea to include single particle ICP-MS measurement in the future.

Answer: *As intuitively suggested by the reviewer, these metal-doped plastics are suitable for measurement by single-particle ICP-MS and one of the authors has already performed a number of studies utilizing single-particle ICP-MS to assess the association of nanoplastics (NPs) with human cell lines⁶ and the extent that NPs pass the gut barrier of fish⁷. Similarly, single particle detection in the context of single-cell mass cytometry for the association of metal-doped NPs to immune cells was also recently published⁸. Although using this approach (compared to bulk ICP-MS analysis, as was done in this present work) would have increased the sensitivity for metal-doped NPs detection, it would have also significantly increased the time for methodological improvements to extract NPs from a variety of tissues. While we agree with the reviewer that the application of single-particle ICP-MS could be of*

benefit in future studies, the purpose of this present study was to compare three different approaches for metal-doped NPs' quantification across a larger number of samples. In this context, bulk ICP-MS analysis was shown to have the highest throughput and lowest detection limit of the methods which were tested.

4. Term of "intermediate" and "long"

Generally, the duration of toxicity tests to evaluate long-term toxicity is 6 months or more. Also, the duration of toxicity tests to evaluate subacute toxicity is 1 to 3 months. Therefore, I think it would be a good idea to revise the names of the administration periods according to the general definition of animal experiments.

Answer: *We thank the reviewer for this thoughtful comment. We adapted the terminology according to toxicity testing and changed to the terms acute (4 hours exposure), subacute (10 days exposure) and subchronic (5 weeks exposure) throughout the entire manuscript. We hope that it is now more applicable.*

5. The experimental term and the displayed term are different.

The intermediate term is a 10-day administration period (Fig. 1), but Fig. 4b only displays up to 9 days. Also, the long term is 5 weeks (Fig. 1), but Fig. 5b only displays up to 10 days.

Answer: *In the intermediate term/subacute exposure experiment feces were collected on alternating days for 24h after gavage, which required putting the animals in specific cages with metal grid floor during that time. As this stresses the animals, which could also affect our measurements, we only selected 4 timepoints for 24h feces collections and decided to only collect feces for 24h after the second last gavage before the final sampling timepoint. Thus, the last measurement cycle of feces was performed on day 9 and not on day 10. Please also see the experimental scheme in Figure 1.*

In the long-term/subchronic exposure experiment, a similar feces analysis setup was used as in the short-term/acute exposure experiment to analyze the excretion of particles for 10 hours after the gavage. However, in this experiment, we used more physiologically relevant concentrations, increased the observed sampling timeframe up to 10 hours and also included a disease mouse model (APCmin+). In order not to stress the animals too long, feces from wildtype and APCmin+ cohorts were only collected for 10 hours, and the animals were then put back into their regular cages for the remaining 14 hours until the final sampling timepoint. Please also find the details in the Methods section about animal experiments (line 715 onwards) and in the experimental schematic presented in Figure 1.

6. Regarding the results of dual counts by IMC (Fig. 3d, 4d, 5e)

It is difficult to distinguish the points of the dot plots shown, and it is unclear in which conditions/organs Pd is accumulating. It would be a good idea to modify the data so that the average and distribution of the data can be visualized by making the Y-axis logarithmic, adding jitter settings to the dot plot, combining the dot plot with a box plot/violin plot, etc.

Also, does this mean that the results show the Pd signal intensity in all pixels in the ROI, and that the data for pixels where Pd is not accumulating is also included? It would be better to clarify whether the results show only data above the LOD.

Answer: Each point in Figures 3d, 4d and 5e represents one pixel and the y-axis gives the intensity of the Pd signal for that specific pixel. As most of the pixels had no Pd detected, they have a signal intensity between 0 and 1 which represents the noise range of the detector. There are very few points above this noise range which are pixels that did contain Pd-NPs. To clarify this, we added to the above mentioned figures a transparent grey square to the region from 0 to 1 to indicate the noise range and better highlight the Pd-NP signal pixels above this area. Since we want to draw the readers' attention only to the points that are above the noise range, we rather not use a logarithmic scale. We have added the following paragraph to the manuscript to clarify this point and thank the reviewer for this helpful comment.

Lines 403-406: Each spot in the pixel counts dot plots (Figures 3d, 4d and 5e) represents one pixel and the y-axis provides the intensity of the Pd signal for that specific pixel in dual counts. As most pixels had no Pd detected, they have a signal intensity between 0 and 1 (in the grey range), which is the noise range of the detector.

7. "dual counts"

I think the term "dual counts" refers to the pulsed mode and analog modes of the detector of ICP-MS, but normally the modes are switched automatically, and signal intensities are provided as count (cps) in both modes. I think it would be sufficient to describe how the authors adjusted for the difference in sensitivity between the pulsed mode and analog mode and write "signal intensity (count)".

Or does Cy-TOF always output both modes signals? If so, it would be better to explain how you processed the signal intensity. For example, the average value of the pulsed and analog modes was used. Or if the pulsed-mode signal exceeded 100 counts, for example, the analog-mode signal was used.

Answer: In this case, the analytical tool we are discussing in this section is IMC and not ICP-MS. The term dual counts stems from the dual scale that is used to get the detection number which is the output of the imaging mass cytometer (IMC). This dual scale considers the number of pulse counts, or number of ions hitting the detector, as well as the intensity of this signal. An IMC does not have the same modes as an ICP-MS and with the pulse count and the intensity, the number of dual counts is extrapolated. If a low number of ions is detected, the dual count is the same as the pulse or ion count and if there is a high abundance signal, the intensity is also used as a more accurate measure. In this study, the dual counts represent the number of ions hitting the detector and an increased intensity means that more than one ion hits the detector at the same time. On lines 923 to 926 in the manuscript, we addressed this in the following text: In mass cytometry, the output of the detector is in dual counts. This is a dual scale encompassing single ion counts for low-level signals and intensity for higher signals. For the purposes of this study, the dual counts detected were low enough to reflect single particles, called pulses, hitting the detector. That is, if one has 10 dual counts of a given metal, this equates to 10 ions of that metal detected.

8. Fig. 3a

Wouldn't it be better if the optical image was also shown?

Answer: We thank the reviewer for this advice and have now added the photographs shown below to the Supplementary Information (Figure S1), where we directly compare the optical images to the reconstructed mass maps.

Figure S1: Direct comparison of reconstructed Pd distributions (upper row) to photographs of measured small intestine samples (bottom row) in the XFI setup.

9. Regarding the IMC tissue images (Fig. 3c, 4c, 5d)

Please provide the units for the scale bars in these figures. Also, please include the color bars and their units. The yellow pixels are difficult to see. Please enlarge them or add arrows so that we can see the specific areas.

Answer: We thank the reviewer for this comment and have added the color bars and units (μm) to the scalebars. The few yellow pixels which represent a positive signal are difficult to see by eye in Figures 4c and 5d and even if we enlarge the image, the signal is very low. Therefore, we decided to use dot plots to show very low signal counts above the background. The yellow pixels are visible in Figure 3c because in this case much more Pd-NPs were detected in the shown regions of interest. To further clarify this for the readers, we have added the following paragraph to better explain our chosen representation.

Lines 399-403: As in Figure 4c, the reader will notice that the yellow pixels containing Pd are impossible to see by eye because they are mixed with the background noise of the detector. They are shown to provide an example of what it looks like when we have extremely few positive pixels for Pd. For this reason, we use the pixel counts dot plots with dual counts to show that these images do in fact contain a couple of pixels with Pd in them (signals above the background of 1 dual count).

10. Fig. 4d

Please specify whether the control shown here is an untreated control or a colitis control.

Answer: The control samples shown in Figure 4d did not receive nanoplastics and did not have colitis. In this subfigure we show samples from the colon, small intestine, kidney and liver from a total of 3 mice which got different treatment: no nanoplastic exposure + no colitis (untreated control), nanoplastic exposure to a healthy mouse, or nanoplastic exposure to a DSS (colitis) mouse model. To specify this more clearly, we exchanged in the figure subscription and throughout the whole text the term "control" with "untreated control".

11. Fig. 5e Insufficient information on the x-axis

For example, Brain1 shows four dot plots. Please explain what is different.

Answer: *The sample "Brain1" shows 4 regions of interest (ROI), each with the dual counts from every pixel. We have added ROI numbers to each section to clarify that we are showing regions of interest and the quantification of the dual counts per pixel in those regions.*

12. Mass numbers are generally written in the left superscripts

Answer: *We apologize for this mistake and thank the reviewer for pointing it out. We have now changed all mass numbers to left superscripts in the whole manuscript.*

13. Regarding the measurement of Pd using ICP-MS

It is said that similar values were obtained by monitoring $m/z = 105, 106,$ and 108 (Line 542), but it is unclear which results were used as the basis for the discussion.

Also, since there is spectral interference on $m/z = 106$ and 108 from 106Cd and 108Cd , it seems appropriate to use the results for $m/z = 105$. Also, was the measurement performed in KED mode? It would be better to clarify the measurement conditions, such as the flow rate of the CRC gas.

Answer: *We have added the following information to the Methods section:*

Lines 811-812: Measurements were made without the use of a collision cell. The flow rate of plasma gas was 15.00 L/min, the flow rate of the carrier gas was 1.02 L/min and the integration time was 0.5 seconds.

Reviewer #3 (Remarks to the Author):

This study proposes a workflow that combines three methods (ICP-MS, XFI, IMC) with using nano-PS particles labeling with Pd to measure the residual NP in test animals. The experimental design using NP inside-including Pd for minimizing influence of labeling on the residual behavior in the body and using three different analytical methods is interesting. However, for the following reasons, the novelty and significance of this proposal for the investigation of the residual behavior of NP have not been sufficiently demonstrated.

1. A clear comparison of the analytical merits of the three methods (ICP-MS, XFI, and IMC) employed in this study should be made for understanding the proposed workflow and selection of order of the methods: 1) between the three methods. For example, detection limit as numbers of NPs as well as Pd content, spatial resolution, and volume or weight concentration, quantification and its reproducibility, and measurement interferences. Comparisons under identical sample conditions are useful if possible. 2) Performance comparison between these three methods and previous methods (e.g., IR, pyrolysis GC/MS, microscopic observation of particles, etc.) 3) comparison of time, and complexity of procedure, cost of the analytical methods.

Answer: *As elaborated in the Introduction, several analytical challenges exist when it comes to the assessment of nanoplastic particles (NPs) in biological samples. To overcome the issues described, we suggest using the combination of XFI, ICP-MS and IMC to quantify metal-doped model NPs. A*

comparison of the main parameters of all three methods has now been added to the Supplementary Information. For a detailed version of the added supplementary Table S7, we kindly ask to refer to our answer for reviewer #1, question 1, or to the Supplementary Information (Table S7) of the manuscript.

2. In the workflow scheme, XFI is proposed as a primary screening method for detecting NP-Pd, however, the application range is not clearly stated. It has a lower detection sensitivity than IMC and ICP-MS, and there is also a possibility that NP-Pd particles will be lost due to pretreatment and cleaning, so it may be unsuitable for primary screening because there is a possibility that residual NP-Pds in rat will be overlooked.

Answer: *As elaborated in the Introduction, we decided to first use non-destructive XFI scans to get an initial impression of the spatial distribution of nanoplastics (NPs) in the samples. The main advantage of using XFI as the first method is that one gets a good first impression of the NP distribution without the need for special sample preparation or even destruction of the samples. It is true that the detection sensitivity of XFI is lower compared to the other methods, but since ICP-MS and IMC are destructive and partly very time-consuming methods, a good knowledge of regions of interest a-priori is required, which can be acquired with XFI. Indeed, it could happen that low particle concentrations could be overlooked when using this workflow, but in terms of time, cost investment, and maintaining sample integrity for subsequent analysis, our proposed scheme provides a good compromise. Also, if detailed IMC analysis or sample processing for further analyses (e.g., immunohistochemistry) is not wanted a-priori, samples can also be simply collected in tubes, air dried or formalin fixed for XFI measurements (like we proposed for stool samples), which avoids potential particle loss.*

3. The NP-Pd particles in the particle solution that are ingested by the rats need to be dispersed and homogenized for the purpose of this study, but this information is not available and cannot be verified. The residual behavior may change depending on whether the NP-Pd particles are dispersed or aggregated.

Answer: *We agree with the reviewer that the behavior of the particles will change with their aggregation state, but we cannot predict the exact state in our exposure scenarios as this depends on different parameters such as the pH value and the concentration and composition of other matter present in the digestive tract of the mice. However, we assured that the particles applied here were initially dispersed, and we sonicated the gavage solution always prior to application. We added the following sentence to the Methods section to more clearly describe our experimental procedures:*

Lines 723-726: Particle solutions for gavage were prepared at respective experimental dilutions in one batch for the whole experiment and aliquots stored at 4°C. Gavage solutions were briefly sonicated before application in mice to ensure particle dispersion.

4. Is the homogeneity of the samples of rat organs and feces after exposure guaranteed between the three methods (ICP-MS, XFI, IMC)? The manuscript mentioned that there is a possibility of loss of NP-Pd particles due to washing operations, etc., but have it been verified the exclusion ratio into the washing solution? Also, the manuscript mentions that the organs were divided into two parts, but how were they divided? If it is the upper and lower parts of the organ, is there a possibility that it will accumulate in one part depending on the type of organ?

Answer: *Since we do not have the techniques to measure small amounts of plastic particles in large volumes, we did not specifically measure the exclusion ratio of particles into the washing solutions. However, given the fact that the results from the XFI measurements, where the samples had been*

processed and analyzed differently compared to those measured with ICP-MS agree very well (see Figures 3-5), we are confident that particle loss did not play a significant role in our study. It is also important to mention again that we only had very few individuals per study, which automatically implies some degree of variation.

Regarding equal organ division, the entire procedure was done in accordance with our animal pathologist Prof. Dr. Lukas Kenner (a co- and corresponding author on this manuscript) in order to ensure that equal parts of the organs were used for the analysis with ICP-MS and XFI (e.g., the brain was cut along the longitudinal fissure separating it into right and left hemisphere). Furthermore, for all organs the total organ weight was measured as well as the partial weight of the ICP-MS part, thus enabling proper recalculation from the partial to the total organ. A strong accumulation in only one organ half would have been detected with XFI but this was not the case – very similar results could be seen with ICP-MS and XFI instead (see Figures 3-5).

5. The reason for setting the amounts of exposure doses for rats is not clearly stated. For example, the dose for the short-term study seems to be high, but is it exceeding the normal range of rat physiology and causing abnormal excretion behavior?

Answer: *We added the following paragraph to state the reason for the different exposure doses more clearly.*

Lines 220-225: Three different concentrations of Pd-NPs were used for the experiments. In the pilot acute exposure scenario, we used the undiluted stock concentration. Although this concentration is high, we needed this high dose of particles to test if we were able to detect the particles with the different analysis methods and to also track the NP uptake into the body as we already expected uptake amounts to be very low after this short exposure period. For the subacute and subchronic exposure we reduced the applied daily dose to more physiological concentrations of 1 mg and 0.5 mg/ day.

6. The superiority of the residual measurement method using PdNP in this study is unclear. It should be compared the behavior of other types of NPs (e.g., only PS, NPs with another tag molecules, metal nanoparticles) in previous studies.

Answer: *The use of model NPs with a conservative tracer can circumvent a number of the analytical challenges associated with detection and quantification of nanoplastics in laboratory experiments. A number of different doping techniques have been put forth in recent years including the addition of fluorescent dyes, inorganic metal-doping, radio-labeling or stable-isotope labeling of the carbon polymer backbone. The commercial availability and ease of use of fluorescently labeled model MNPs have made them a popular choice amongst researchers, but challenges associated with leaching of dyes, as well as quenching of the fluorescence with organic matter, have caused concern in the past. Radio or stable isotope labeling offers excellent detection limits, but the analytical equipment needed in order to measure the materials is often not commonplace. Consequently, metal-doping has several key advantages, since existing standard methods for trace metal analysis exist and can be exploited for measuring metal-doped plastic materials. These techniques also have excellent detection limits, and when the metal-dopant is a rare metal and the materials are designed in a way that no leaching occurs, it is a stable and unique signature to measure in the experimental matrix.^{9,10}*

We have added the following paragraph to better explain why we have chosen to use model metal-doped nanoplastics:

Lines 74- 89: When choosing model MNPs to study, one must keep in mind that MNPs are a large suite of contaminants with different properties such as polymer type, surface chemistry, extent of weathering and others. These different physiochemical properties may also lead to variations in uptake and toxicological impacts, but the extent of this variation is not yet known and is currently under investigation by many scientists in the field. Therefore, we have chosen to use model nanoplastics with defined properties administered as a homogeneous suspension. The model metal-doped materials are a good proxy for environmental nanoplastics due to their surface composition and irregular surface. A number of different doping techniques have been put forth in recent years including the addition of fluorescent dyes, inorganic metal-doping, radio-labeling or stable-isotope labeling of the carbon polymer backbone. The commercial availability and ease of use of fluorescently labeled model MNPs have made them a popular choice amongst researchers, but challenges associated with leaching of dyes, as well as quenching of the fluorescence with organic matter, have caused concern in the past. Radio or stable isotope labeling offers excellent detection limits, but the analytical equipment needed in order to measure the materials is often not commonplace. Consequently, metal-doping has several key advantages, since existing standard methods for trace metal analysis exist and can be exploited for measuring metal-doped plastic materials.^{9,10}

7. Rat experiments:

Did the rats consume the entire amount of the garbage solution? In the high-dose condition, did the rats not immediately vomit due to a sense of foreign matter? Also, are multiple rats kept in one test cage? If so, is there a possibility of cross-contamination due to accidental ingestion of feces from other rats?

Answer: *The gavage solution was applied as 100 µL with a clean metal feeding needle directly into the stomach. This is a common procedure used in animal experiments and 100 µL is an adequate volume for a mouse. Also, worth mentioning, mice do not have a gag reflex and are indeed physiologically incapable of vomiting. While there is the possibility of cross-contamination or “extra dosing”, as mice sometimes tend to eat feces (their own as well as that of other individuals), this cannot be avoided unless the mice are constantly kept on a metal grid floor which in turn would cause constant stress and therefore is ethically not responsible for longer timeframes. However, as seen in our measurements, not all feces contained Pd-NPs and only the ones excreted at 3-5 hours after gavage (for healthy mice) showed increased concentrations, but at maximum 12.5% of the daily dose. Thus, the chances of significant “cross contamination” due to mice eating feces are rather limited.*

8. Condition of ICP-MS instruments at the ICP-MS and IMC methods should be added: the conditions of ICP-MS instruments such as flow rates of plasma and carrier gas, and integration time. the condition for IMC such as wavelength and pulse speed of ablation laser, sample scanning speed.

Answer: *We have added the following information to the Methods section:*

Lines 811-812: Measurements were made without the use of a collision cell. The flow rate of plasma gas was 15.00 L/min, the flow rate of the carrier gas was 1.02 L/min and the integration time was 0.5 seconds.

Lines 874 onwards: In IMC, a tissue slide is inserted into the Hyperion™ Tissue Imager (HTI, Standard BioTools) where a UV laser ablates the tissue in consecutive 1 µm diameter shots (Figure S5). A new MCD file was created, and a slide image (taken with a mobile phone) was aligned with the camera view image in the CyTOF software (version 7.0.8493) so that the laser position indicates the same region in both images. All ROI were ablated at 200 Hz laser frequency and 3 decibels (db) laser power with a scan area timing of $\geq 1\text{mm}^2/2$ hours.

9. L576-587, XFI measurement:

Is the depth that can be measured equivalent to the thickness of the sample? If it is only the surface, is the sample's inhomogeneity not affected? Also, to check the degree of a strong correlation between reconstructed and anticipated values from ICP-MS measurements, please show specific data.

Answer: *As hard X-ray photons of 53 keV energy were used for the XFI measurements, the penetration depth of the incident and emitted fluorescence photons is larger than the thickness of the samples. However, there is some degree of attenuation which has to be considered when reconstructing quantitative mass values (details please see lines 846 onwards in the Methods section). Potential inhomogeneities of the samples' surfaces do not have an influence on XFI data. We compare the quantitative results of our measurements of the gavage solution with ICP-MS and XFI at the beginning of the Results section in the manuscript (lines 233 onwards) and show further direct comparisons in Figures 3-5.*

10. Table S7:

Since it is a group of particles, the hydrodynamic diameter should show the distribution, not just the average value.

Answer: *The z-average hydrodynamic diameter (d_{zH}) which we have provided in Table S8 follows a Gaussian (Normal) distribution, also described by the standard deviation value. We agree that showing the full distribution would be more appropriate if the nanoplastics suspension was polydisperse, however our nanoplastics are monodisperse (polydispersity index = 0.1).*

References:

1. Stock, V. *et al.* Uptake and effects of orally ingested polystyrene microplastic particles in vitro and in vivo. *Arch Toxicol* **93**, 1817–1833 (2019).
2. Carr, K. E., Smyth, S. H., McCullough, M. T., Morris, J. F. & Moyes, S. M. Morphological aspects of interactions between microparticles and mammalian cells: intestinal uptake and onward movement. *Progress in Histochemistry and Cytochemistry* **46**, 185–252 (2012).
3. Schmidt, C. *et al.* Nano- and microscaled particles for drug targeting to inflamed intestinal mucosa—A first in vivo study in human patients. *Journal of Controlled Release* **165**, 139–145 (2013).
4. Yoo, J.-W., Doshi, N. & Mitragotri, S. Adaptive micro and nanoparticles: Temporal control over carrier properties to facilitate drug delivery. *Advanced Drug Delivery Reviews* **63**, 1247–1256 (2011).
5. Pradel, A. *et al.* Stabilization of Fragmental Polystyrene Nanoplastic by Natural Organic Matter: Insight into Mechanisms. *ACS EST Water* **1**, 1198–1208 (2021).

6. Hendriks, L., Kissling, V. M., Buerki-Thurnherr, T. & Mitrano, D. M. Development of single-cell ICP-TOFMS to measure nanoplastics association with human cells. *Environ. Sci.: Nano* **10**, 3439–3449 (2023).
7. Clark, N. J., Khan, F. R., Mitrano, D. M., Boyle, D. & Thompson, R. C. Demonstrating the translocation of nanoplastics across the fish intestine using palladium-doped polystyrene in a salmon gut-sac. *Environment International* **159**, 106994 (2022).
8. Fusco, L. *et al.* Nanoplastics: Immune Impact, Detection, and Internalization after Human Blood Exposure by Single-Cell Mass Cytometry. *Advanced Materials* 2413413 (2024) doi:10.1002/adma.202413413.
9. Mitrano, D. M. *et al.* Balancing New Approaches and Harmonized Techniques in Nano- and Microplastics Research. *ACS Sustainable Chem. Eng.* **11**, 8702–8705 (2023).
10. Abdolahpur Monikh, F. *et al.* The analytical quest for sub-micron plastics in biological matrices. *Nano Today* **41**, 101296 (2021).

Reviewer #1:

I think the expression “in the digestive tract” is the cause of the misunderstanding. Does this mean that the particles penetrate the intestinal barrier and were detected in the tissue, but rather that they were detected in a mixture of contents and tissue? I was unable to determine whether it was in the lumen or tissue.

If the particles passed through the intestinal barrier, it may be in the blood. Furthermore, even if the particles were not detected in the blood at a certain time point, it cannot be denied that this is the result of rapid distribution to organs. Therefore, I thought that tracking the kinetics in the blood over time and calculating the elimination half-life, etc., would be useful in solving this question. Moreover, if the particles that passed through the intestinal barrier are excreted in the feces, it is possible that they were transferred to the feces via bile excretion from the liver.

Answer: We understand how our choice of terminology is misleading, as “digestive tract” includes the organs tissue and the contents in the lumen. We therefore changed to “in the lumen of the digestive tract” where appropriate, in the manuscript to make this clearer.

IMC analysis of acute exposure in the small intestine (see Figure 3c) showed that the majority of Pd-NPs was detected in the lumen, but also some positive pixels were detected in the epithelial tissue, thus indicating some (minor) uptake across the intestinal barrier, and probably transport via blood to several organs. We agree that assessing Pd-NP half-life in the blood would be a very interesting point in terms of uptake kinetics, however, against our own expectations we were not able to detect or quantify Pd-NPs via XFI or ICP-MS in the blood samples of the acutely exposed mice (please see also Supplementary Table 2C), except for a single sample. Thus, we suggest that Pd-NP concentrations in the blood might be lower than the detection limits of our methods and prospectively larger volumes of blood (> 100 µL) would be needed to detect and quantify Pd-NP contents in the blood.

Reviewer #2:

The authors adequately responded to the reviewers’ questions. Most of my questions have been resolved. However, I think the following corrections are necessary.

1. Model animal

The authors used an animal model of small intestinal disease. I want the authors to explain in the introduction why they focused on the small intestine and the relationship between the disease and NP.

Answer: To clarify the use of the two disease models we added the following paragraph in the introduction to state this more clearly:

Line 119 ff: “Colitis is an inflammatory process, which mainly affects the colon and causes a leaky intestinal barrier which is supposed to lead to higher MNP uptake rates into the body⁴⁷.”

The APCmin+ model, in contrast, is a genetic model of early-stage intestinal neoplasia, with benign adenomatous polyps mainly forming in the small intestine and to a lesser extent in the colon^{48,49}. The model is considered similar to human familial adenomatous polyposis (FAP) and also shows to be associated with a defective intestinal barrier and increased permeability⁵⁰. As the GI tract is chronically exposed to orally ingested MNPs and - aside from inhalation – probably one of the main entrance routes for MNPs into the body, we also wanted to investigate Pd-NP uptake under these pathological conditions.”

2. Fig 5e
“ROI 4-7” should be “ROI 5-7”

Answer: We thank the reviewer for pointing out this error, and we have now corrected Figure 5e. In addition, we added further information about the mice samples which are shown in the respective figures to the supplementary material (Figure S3).

Reviewer #3:

I would like to appreciate for your response and revision to the reviewer’s comments. However, a few questions regarding some parts of your response are remained. I would like to recommend to clarify the following points and revision of the manuscript.

To Answer 1.

At the columns about reproducibility in Table S7, specific examples of numerical data should be added to clarify the comparison.

Answer: We have now added the following numerical data examples for the three methods to Table S7.

“The average RSD for triplicate measurements of nanoplastics was 0.88 +/- 0.45 %.”

“From all our synchrotron-based XFI measurements we have determined a reproducibility of 5% in terms of the fluctuations in counts of the measured photons. At the beginning of each measurement day we perform a photon flux reconstruction, typically with errors below 5%.”

“We limit our ROI size to between 1-2 mm². The short exposure experiment in this study had 24 ROI acquired and an acquisition time of two days.”

“IMC is highly reproducible because metals are more stable than fluorophores and do not degrade over time. Also, laser ablation is quality controlled daily to ensure correct ablation and sensitivity.”

To Answer 2.

Regarding XFI, the answer states that XFI is non-destructive and does not require special sample treatment. While the embedding treatment is correct for tissue preservation, there is no information provided to determine whether the preservation of nanoparticles remained at organs is acceptable, making it impossible to judge. The embedding treatment used isopropanol and additionally paraformaldehyde solution for the small intestine and colon. At

the very least, the organs have been in contact with these solutions, and the possibility of particle loss cannot be excluded. As requested in the original comments from the reviewers, the effects of particle loss due to contact with these solutions should be discussed and the revision of the manuscript is recommended. Furthermore, although the limitations due to lower detection sensitivity of XFI compared to other methods are mentioned in the answer, this should also be clearly stated in the manuscript. In addition, Table S7 has been added, so it is recommended that the advantages and disadvantages of XFI, ICP-MS, and IMC are discussed at each item (e.g., time, cost, and reproducibility).

Answer: We thank the reviewer again for the valuable feedback and have now added the following information to the manuscript and the supplementary information:

Lines 130 – 132: “Such scans do not require any special sample preparation and provide a first quantitative impression of the dynamics of Pd-NPs, however, the achievable sensitivity is lower compared to the other methods.”

Regarding XFI, the following information was added to Table S7:

“Synchrotron for high-sensitivity measurements is needed, but compact, much less expensive systems are under development.”

Regarding IMC, the following information was added to Table S7:

“IMC directly measures metal isotope counts based on the time the metal hits the detector. This is extremely precise because this time is verified daily in the quality control methods.”

Furthermore, a thorough discussion about the advantages and disadvantages of all three methods can be found in the Discussion of the main manuscript (especially lines 457 to 470).

We observed good correlation of ICP-MS (minimally processed samples) and XFI (FFPE processed samples) measurements, especially for the acute exposure samples, pinpointing at only minor losses of Pd-NPs during tissue FFPE processing. We elaborated this point in more detail in the respective paragraph at the end of the Results section (changes highlighted in blue):

Lines 457-470: “In contrast, sample preparation of the organ halves for ICP-MS measurements only required drying of the native organs and transferring the samples from transport tubes to digestion tubes, to assure for a minimal loss of Pd-NPs from sampling to analysis. For XFI and IMC measurements, however, the organs were fixed and desiccated, requiring several washes in different solutions which may lead to washing out some of the particles. Although XFI would be also able to measure Pd-NP content in dried organs (similar to ICP-MS, without sample processing to avoid potential particle loss), we proposed the workflow with paraffin embedding for better correlation with IMC analysis and to further enable subsequent immunohistochemical analysis if desired. Chemical compatibility of polystyrene particles with fixation and a desiccation protocol was already described by Gonçalves et al.⁵¹ and independently verified a priori (data not shown). Furthermore, good correlations between values obtained by ICP-MS, where samples were only dried, and by XFI, where samples were processed with FFPE, especially for acute exposure samples, highlight the negligible losses of Pd-NPs during tissue FFPE processing.”

To Answer 3.

Although an operation to promote particle dispersion has been clearly added, it has not been confirmed whether the particles were dispersed. I recommend evidence of the dispersion is added for the confirmation.

Answer: The laboratory that performed the mice exposures was not equipped with a DLS. Therefore, it was not possible to measure dispersion directly before administering the particles. However, this dispersion protocol enabled us to repeatedly measure the NP initial size and polydispersity in the laboratory equipped with DLS (ETH Zürich).

To Answer 4.

It claims that the agreement between the XFI and ICP-MS measurements indicated that particle loss due to washing is minimal. However, if the influence of leaching due to pretreatments in both XFI and ICP-MS is significant, and even under conditions where the extent of influence is comparable, the agreement between XFI and ICP-MS analysis results still hold. Therefore, returning to the original reviewer's comment, it is necessary to evaluate or estimate the particle loss effect of each pretreatment of XFI or ICP-MS under the experimental condition.

Answer: We want to highlight that for ICP-MS the samples were minimally processed, only requiring the samples to be put in transport tubes (feces) or for organ(halves) dried on filter paper before being shipped to the laboratory which performed the analysis for particle quantification. The whole organ+filter paper or feces was then transferred from the transport tube to a specific digestion tube and digested as a whole, thus ensuring only minimal Pd-NP loss during this procedure. As we see good correlation with minimally processed ICP-MS samples and FFPE processed XFI samples, we suggest that only minor amounts of Pd-NPs might be lost during the tissue processing.

We elaborated this point in more detail in the respective paragraph at the end of the Results section (see also our comments from Answer 2):

Lines 457-470: "In contrast, sample preparation of the organ halves for ICP-MS measurements only required drying of the native organs and transferring the samples from transport tubes to digestion tubes, to assure for a minimal loss of Pd-NPs from sampling to analysis. For XFI and IMC measurements, however, the organs were fixed and desiccated, requiring several washes in different solutions which may lead to washing out some of the particles. Although XFI would be also able to measure Pd-NP content in dried organs (similar to ICP-MS, without sample processing to avoid potential particle loss), we proposed the workflow with paraffin embedding, for better correlation with IMC analysis and to further enable subsequent immunohistochemical analysis if wanted. Chemical compatibility of polystyrene particles with fixation and a desiccation protocol was already described by Gonçalves et al.⁵¹ and independently verified a priori (data not shown). Furthermore, good correlation of ICP-MS (minimally processed samples) and XFI (FFPE processed samples) measures, especially for the acute exposure samples, pinpoint at only minor losses of Pd-NP s during tissue FFPE processing."

To Answer 6.

Thank you for adding examples of conventional labeled NPs. However, please cite each reference for labeling with fluorescent dyes, inorganic metals, radioactive substances, or stable isotopes of carbon polymers for understanding.

Answer: Thank you for this important comment, we have now added the following four references to the manuscript (lines 79-80):

[29] Catarino, A. I., Frutos, A. & Henry, T. B. Use of fluorescent-labelled nanoplastics (NPs) to demonstrate NP absorption is inconclusive without adequate controls. *Science of The Total Environment* 670, 915–920 (2019)

[30] Cassano, D., La Spina, R., Ponti, J., Bianchi, I. & Gilliland, D. Inorganic Species-Doped Polypropylene Nanoparticles for Multifunctional Detection. *ACS Appl. Nano Mater.* 4, 1551–1557 (2021).

[27] Al-Sid-Cheikh, M. *et al.* Synthesis of ¹⁴C-labelled polystyrene nanoplastics for environmental studies. *Commun Mater* 1, 97 (2020)

[12] Mitrano, D.M., Wick, P. & Nowack, B. Placing nanoplastics in the context of global plastic pollution. *Nat. Nanotechnol.* 16, 491-500 (2021).

To Answer 8.

Information on laser ablation has been added, but important laser wavelengths are missing. In addition, laser output should be normalized per area and expressed in units such as J/cm².

Answer: We have adapted line 673 (changes highlighted in blue) to add further important information as follows:

"In IMC, a tissue slide is inserted into the Hyperion™ Tissue Imager (HTI, Standard BioTools) where a 213-219 nm UV laser ablates the tissue at a frequency of 200 Hz in consecutive 1 μm diameter shots (Figure S6)."

The energy output per unit area is not published by Standard BioTools to the best of our knowledge, but an approximate value can be extrapolated based on the spot size, pulse energy and repetition rate.